# IntDiff: Mitigating reward hacking via Diffusion-specific Intrinsic rewards for Diffusion model fine-tuning

## Abstract

Diffusion models show excellent generative power but remain weak in aligning with specific tasks. RL fine-tuning improves alignment yet introduces three major drawbacks: (1) Rewards are evaluated only on the final image, so intermediate steps lack feedback, leading to reward hacking and distributional deviation; (2) In diffusion models, RL's inherent exploration–exploitation imbalance persists and is amplified by the long sequential denoising; (3) Current RL methods train all samples with fixed steps, wasting compute. To address the above issues, this paper proposes IntDiff. First, *a diffusion-specific intrinsic reward paradigm* is proposed, which offers intermediate exploration to mitigate sparse rewards. Second, a filtering mechanism is introduced to penalize exploration that reduces text-image consistency. Third, we develop an adaptive method that uses denoising progress to balance exploration and exploitation: denoising Intrinsic Reward drives diverse exploration early, then yields to target reward maximization later. Furthermore, we use dynamic early stopping that adjusts training steps to prompt difficulty, saving computation. *Finally, we theoretically establish the soundness of the denoising intrinsic reward.* Experiments show that compared to the existing *best* baselines, IntDiff achieves maximum improvements of diversity by 60% while reducing computational cost by 22%.

## 1 Introduction

Diffusion models (Ho et al., 2020; Nichol et al., 2021; Ramesh et al., 2022; Rombach et al., 2022; Saharia et al., 2022) have become one of the most dominant paradigms for text-to-image generation in recent years. Benefiting from the step-by-step denoising formulation, they demonstrate significant advantages in restoring image details and producing visually natural outputs. However, diffusion models are trained with maximum likelihood (Ho et al., 2020; Black et al., 2023) to reproduce training data distributions rather than optimize for specific tasks. Thus, they remain insufficient in directly aligning with downstream objectives, which severely limits their practical adoption (Franceschelli & Musolesi, 2024). Recent studies leverage reinforcement learning (RL) to optimize diffusion models for specific goals such as aesthetic quality. These methods model the denoising process as a Markov Decision Process (MDP) (Bellman, 1957; Puterman, 1990), and apply task-directed rewards to the final results to guide the policy towards specific downstream objectives (Black et al., 2023; Fan et al., 2024; Yang et al., 2024a; Wallace et al., 2024; Wu et al., 2023b; Chen et al., 2024; Jia et al., 2025).

Although RL methods have improved task adaptation, they still face three significant challenges. (1) Rewards only on the final image, while the intermediate deoising steps lack feedback. This sparse setting drives overfitting to the final reward, which degrades fidelity and diversity(Fan et al., 2024); (2) Exploration–exploitation imbalance is inherent in RL: excessive exploration reduces efficiency, while overemphasizing exploitation risks local optima. In diffusion models, this imbalance is amplified by the long sequential denoising process (Franceschelli & Musolesi, 2024); (3) Current methods set a fixed step size for training, executing uniformly across prompts, and thus waste computation. *Appendix. A provides a detailed discussion of **Related Work** and how **our method differs from existing approaches**.*

To address above challenges, we propose IntDiff, an RL framework that fine-tunes diffusion models to improve generation-trajectory exploration and task alignment under limited compute. IntDiff involves three core aspects:

First, we propose *a diffusion-specific intrinsic reward paradigm (distinct from state-exploration rewards in traditional RL; see Appendix. A.5)* during denoising to mitigate overfitting caused by reward hacking. (Fan et al., 2023). The exploration signals are derived from existing diffusion variables without extra models. Second, to further guide exploration, Semantic Consistency Regularization tracks alignment, penalizes degradation, and drops low-efficiency samples. *After introducing this regularization, IntDiff's exploration behavior is not blindly expanded but instead becomes more reasonable and controlled under semantic constraints.* Third, an Adaptive Coordination mechanism leverages denoising progress to adjust exploration and exploitation. In the early stages, exploration is stressed for diversity, while later stages focus on exploitation for higher target rewards. Furthermore, by continuously estimating denoising progress, we enable dynamic stopping of training steps, replacing the current one-size-fits-all strategy that applies a fixed number of steps to all prompts. Finally, we theoretically demonstrate the rationale for introducing the intrinsic reward mechanism.

Our experiments show improved diversity and text-image consistency, with task rewards significantly increased under mitigated reward hacking. In summary, we highlight three key contributions:

- We propose a diffusion-specific intrinsic reward paradigm that injects exploration signals into denoising steps, thereby alleviating overfitting to sparse target rewards. This is justified under the optimal policy, and *Theorem 2.1 confirms its soundness*.
- We introduce Semantic Consistency Regularization, which tracks exploration, penalizes semantic degradation, and prunes low-efficiency paths to enhance exploration quality and reduce computational waste.
- We introduce an Adaptive Coordination mechanism, which balances exploration and exploitation to address the amplified imbalance in diffusion fine-tuning. It further dynamically truncates training steps, enhancing training efficiency.

## 2 METHOD

This section presents the construction of IntDiff. First, in Sec. 2.2, we design intrinsic rewards based on intermediate denoising states of diffusion models, providing missing feedback signals and encouraging diversity through exploration. Then, in Sec. 2.3, we introduce a semantic consistency filtering mechanism to remove trajectories that degrade text-image alignment, thereby enhancing exploration quality and efficiency. Finally, in Sec. 2.4, we develop a denoising-completeness-based adaptive mechanism that dynamically balances exploration and exploitation, while truncating redundant training steps under adaptive control to save computation. In addition to these targeted innovations, we further replace PPO with a reflective optimization algorithm for policy learning in diffusion models; the details and benefits are discussed in Appendix B.1.

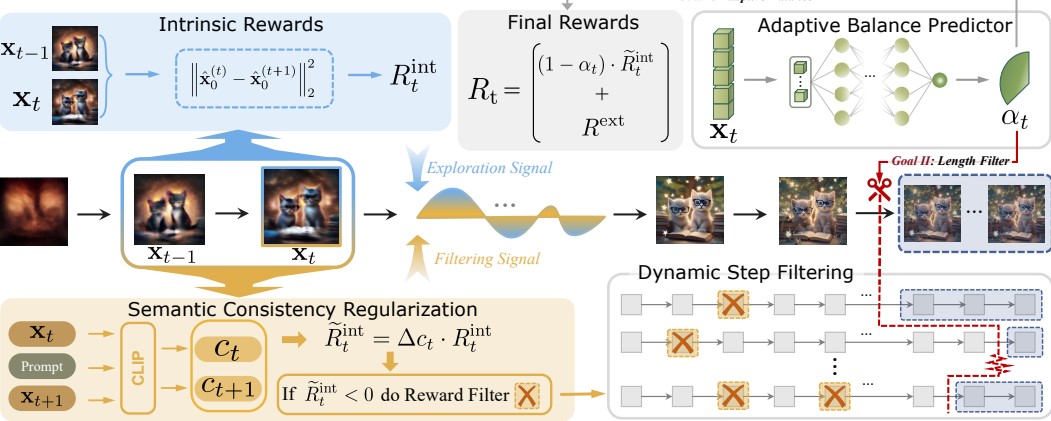

Figure 1: The overall framework of the proposed IntDiff.

## 2.1 EXTRINSIC AND INTRINSIC REWARDS FOR DIFFUSION FINE-TUNING

**Diffusion Models.** A diffusion model characterizes the target data distribution by learning to reverse a predetermined noising process. Starting from a clean data point $\mathbf{x}_0$, the forward diffusion process $q(\mathbf{x}_t \mid \mathbf{x}_{t-1})$ progressively injects Gaussian noise, generating a latent trajectory $\mathbf{x}_0 \to \cdots \to \mathbf{x}_T$, with the terminal state $\mathbf{x}_T$ following $p(\mathbf{x}_T) = \mathcal{N}(\mathbf{0}, \mathbf{I})$. The generative model is trained to learn the reverse-time conditionals

$$p_\theta(\mathbf{x}_{0:T}) = p(\mathbf{x}_T) \prod_{t=1}^{T} p_\theta(\mathbf{x}_{t-1} \mid \mathbf{x}_t),$$

which are trained to match the true reverse posterior $q(\mathbf{x}_{t-1} \mid \mathbf{x}_t, \mathbf{x}_0)$. Sampling begins from noise $\mathbf{x}_T$ and repeatedly applies the learned reverse transitions to reconstruct a data sample. Let

$$q\left(\mathbf{x}_t \mid \mathbf{x}_{t-1}\right) := \mathcal{N}\left(\mathbf{x}_t; \sqrt{1 - \beta_t}\,\mathbf{x}_{t-1},\, \beta_t \mathbf{I}\right),$$

where

$$\zeta_t := 1 - \beta_t, \qquad \bar{\zeta}_t := \prod_{k=1}^{t} \zeta_k, \quad (\bar{\zeta}_0 := 1).$$

Assuming $\boldsymbol{\epsilon}_\theta(\mathbf{x}_t, t)$ denotes the learned noise-prediction function, the associated estimate $\hat{\mathbf{x}}_0$ can then be computed as

$$\hat{\mathbf{x}}_0(\mathbf{x}_t, t) = \frac{1}{\sqrt{\bar{\zeta}_t}} \left(\mathbf{x}_t - \sqrt{1 - \bar{\zeta}_t}\,\boldsymbol{\epsilon}_\theta(\mathbf{x}_t, t)\right). \tag{1}$$

**IntDiff's MDP Formulation.** We formulate the denoising process as a Markov Decision Process (MDP) Puterman (1990); Gan et al. (2024). In this paper, we represent the Markov tuple for the reinforcement learning formulation as $\langle \mathcal{S}, \mathcal{A}, R \rangle$. Here, $\mathcal{S}$ denotes the state space, $\mathcal{A}$ denotes the action space, and $R$ represents the reward function in reinforcement learning. In this paper, $R$ is equivalent to the extrinsic reward. The prompt $\mathbf{z}$ following the distribution $p(\mathbf{z})$. For clarity in the subsequent sections of this paper, we re-denote the number of reverse denoising steps as $t$, that is, $t \leftarrow T - t$. Therefore, in this paper, the state $\mathbf{s}_t$ in reinforcement learning corresponds to the denoising state $\mathbf{x}_t$, and the action $\mathbf{a}_t$ corresponds to $\mathbf{x}_{t-1}$.

In RL fine-tuning, most methods adopt a terminal-only reward: the agent receives a reward $r(\mathbf{x}_0, \mathbf{z})$ only at $t = 0$, while the reward for all other time steps is exactly zero. Producing the Q function $Q^\pi = \mathbb{E}_{\tau \sim p(\tau|\pi)} \sum_t R(\mathbf{s}_t, \mathbf{a}_t)$, where $\pi$ denotes the denoising policy and $\tau$ denotes the denoising trajectory. This sparse feedback leads to training inefficiency and structural or semantic drift, due to the lack of stepwise guidance.

**Extrinsic and Intrinsic Rewards.** To address the above sparse reward issue, we define the Extrinsic and Intrinsic Rewards as follows:

- Extrinsic Rewards($R^{\text{ext}}$): In current RL paradigms for diffusion models, extrinsic rewards refer to metrics assessed on the final image after denoising. Accordingly, we define $R^{\text{ext}} = r(\mathbf{x}_0, \mathbf{z})$, from which it is clear that $R^{\text{ext}}$ is extremely sparse.

- Intrinsic Rewards($\widetilde{R}_t^{\text{int}}$): Given that the sparsity of $R^{\text{ext}}$ is insufficient to support the regular RL training of diffusion models, this study constructs intrinsic Rewards $\widetilde{R}_t^{\text{int}}$ to compensate for the inadequate feedback and improve the optimization process.

In this work, extrinsic rewards correspond to exploitation and intrinsic rewards to exploration. The total reward is:

$$R^{\text{total}} = \underbrace{R^{\text{ext}}}_{\text{exploitation reward}} + \lambda \cdot \underbrace{\widetilde{R}_t^{\text{int}}}_{\text{exploration reward}}. \tag{2}$$

In non-RL diffusion fine-tuning domains $\lambda$ is commonly a fixed weight (Pathak et al., 2017; Burda et al., 2018; Yan et al., 2023). In our work, we replace $\lambda$ with an adaptive factor, as detailed in Sec. 2.4. *Further details on the intrinsic and extrinsic rewards can be found in Appendix A.4.*

## 2.2 Intrinsic Rewards for Diffusion Model Fine-Tuning

Sparse rewards in RL fine-tuning often lead to insufficient signals and overfitting. To mitigate this, we propose an intrinsic mechanism derived from the denoising process to provide additional guidance. Specifically, at each time step, we use Eq. 1 predicts the final image results $\hat{\mathbf{x}}_0^{(t)}$ and $\hat{\mathbf{x}}_0^{(t+1)}$ based on the current state $\mathbf{x}_t$ and the previous state $\mathbf{x}_{t+1}$, respectively. *The intrinsic reward is first defined as the Euclidean distance between them, serving as a **preliminary form** prior to semantic modulation:*

$$R_t^{\text{int}} = \left\| \hat{\mathbf{x}}_0^{(t)} - \hat{\mathbf{x}}_0^{(t+1)} \right\|_2^2. \tag{3}$$

This design provides crucial intermediate feedback by directly measuring the distributional perturbation between consecutive states, thereby reducing the optimization bias inherent in relying solely on terminal extrinsic rewards.

In text-to-image generation, as long as the exploration does not deviate from the prompt semantics, a higher intrinsic reward encourages more active exploration, thereby increasing the diversity of sampling trajectories. Once the exploration departs from the semantic constraint, the intrinsic reward immediately becomes a penalty (i.e., negative reward); the specific design ensuring semantic consistency is provided in Sec. 2.3. *Our intrinsic reward mechanism is tailored to diffusion models and is constructed entirely from variables inherent to the denoising process without any auxiliary modules.*

## 2.3 Semantic Consistency Regularization

Reward hacking in RL fine-tuning of diffusion models manifests mainly in two types of distortion: (1) the generated images tend to converge in structural layout and background composition, resulting in diminished diversity; (2) the model overly conforms to the reward objective, resulting in severe misalignment between images and textual prompts. The former can be mitigated by the intrinsic reward mechanism introduced in Subsection 2.2, which enhances trajectory diversity through intermediate perturbations. However, the latter concerns semantic stability and cannot be effectively constrained by exploration-driven intrinsic rewards alone.

To address the second issue, we propose Semantic Consistency Regularization, which suppresses semantic degradation while ensuring effective exploration. This mechanism calculates the alignment score between the generated image $\hat{\mathbf{x}}_0^{(t)}$ at step $t$ and the textual prompt $\mathbf{z}$ using a frozen vision-language model $F$ (e.g., CLIP):

$$c_t = F(\hat{\mathbf{x}}_0^{(t)}, \mathbf{z}), \quad \Delta c_t = c_t - c_{t+1}. \tag{4}$$

We combine $\Delta c_t$ with the intrinsic reward to construct the final semantically modulated reward signal:

$$\widetilde{R}_t^{\text{int}} = \Delta c_t \cdot R_t^{\text{int}} = \Delta c_t \cdot \left\| \hat{\mathbf{x}}_0^{(t)} - \hat{\mathbf{x}}_0^{(t+1)} \right\|_2^2. \tag{5}$$

By using $\Delta c_t$, we can distinguish the quality of exploration. Specifically:

- $\Delta c_t > 0$: Exploration moves toward improving text–image consistency, $\widetilde{R}_t^{\text{int}}$ gives positive reward to encourage the current exploratory action.

- $\Delta c_t < 0$: Exploration decreases text–image consistency, $\Delta c_t < 0$, so $\widetilde{R}_t^{\text{int}} < 0$. In this case, the intrinsic reward serves as a penalty to suppress the current exploratory action.

If $\Delta c_t < 0$ persists over multiple steps, the current trajectory is regarded an inefficient exploration, and a portion of its RL updates is randomly dropped to reduce the noise of the gradient estimation and save computation, which we termed ***Reward Filter***.

Semantic Consistency Regularization $\Delta c_t$ has three advantages: (1) it dynamically perceives the evolution of semantic consistency and accurately identifies degraded paths; (2) it enables controlled exploration in the semantic space, improving sample quality and update efficiency; (3) it forms a synergistic optimization loop with the explore reward: $\Delta c_t$ provides directional guidance for exploration, while the resulting exploration indirectly enhances text–image alignment (semantic consistency). The two components jointly reinforce each other.

### 2.4 ADAPTIVE TRADE-OFF BETWEEN EXPLORATION AND EXPLOITATION

The exploration–exploitation imbalance has long existed in RL and remains unresolved (Burda et al., 2018; Taiga et al., 2021). In diffusion model fine-tuning, it is further amplified by long sequential denoising (Franceschelli & Musolesi, 2024). This issue is critical: excessive exploration lowers efficiency, while excessive exploitation risks local optima. In other non-RL diffusion settings, some methods use fixed ratios of exploration and exploitation, but such static schemes cannot adapt to changing training dynamics. *To our knowledge, this is the first work to adaptively coordinate intrinsic (exploration) and extrinsic (exploitation) rewards in diffusion models.*

In detail, a perception model $f(\mathbf{x}_t)$ is designed to implement the adaptive trade-off. The input is the intermediate image state $\mathbf{x}_t$ and the output is the estimated denoising completion level $\alpha_t \in [0, 1]$, which serves as a dynamic factor for reward balance and step control. During training, the initial denoised image $\mathbf{x}_T$ is labeled as a 'noisy' sample ($y_T = 0$), and the final image $\mathbf{x}_0$ is labeled as a 'clean' sample ($y_0 = 1$). The perception model is trained using the following cross-entropy loss:

$$\mathcal{L}_{\text{percep}} = -\sum_i \left( y_i \log f(\mathbf{x}_i) + (1 - y_i) \log(1 - f(\mathbf{x}_i)) \right). \tag{6}$$

During the diffusion model training, $\alpha_t = f(\mathbf{x}_t)$ is used as an adaptive factor based on denoising progress, applied in two aspects. First, it is used for adaptive fusion of exploration and exploitation rewards. Therefore, the total reward is updated from Eq. 2 as follows:

$$R^{\text{total}} = \underbrace{R^{\text{ext}}}_{\text{exploitation reward}} + (1 - \alpha_t) \cdot \underbrace{\widetilde{R}_t^{\text{int}}}_{\text{exploration reward}}. \tag{7}$$

Second, adaptive factor is used for **Length Filter** to reduce the length of training steps: based on the generation progress corresponding to the prompt, the number of diffusion steps is dynamically adjusted. When $\alpha_t = 1$ is reached, the subsequent process is terminated early to enable personalized step control and effectively reduce unnecessary computational cost. Under a unified perception signal framework, this adaptive mechanism integrates reward control and computational scheduling, improving training stability, generation diversity, and efficiency.

*Theory of Adaptive Intrinsic Rewards:* *To demonstrate the validity of adaptive intrinsic rewards, we provide the following derivation:*

**Theorem 2.1.** *Assuming $||(1 - \alpha_t) \cdot \widetilde{R}_t^{int}|| \leq \frac{Q(\mathbf{s}_t, \mathbf{a}_t^{opt}) - Q(\mathbf{s}_t, \mathbf{a}_t^{sub})}{2(T-t)}$, for $t = 1, 2, \cdots, T - 1$, where $\mathbf{a}_t^{opt}$ and $\mathbf{a}_t^{sub}$ are the optimal and suboptimal actions respectively, the new MDP $\langle \mathcal{S}, \mathcal{A}, \widetilde{R}^{int}, R_{ext} \rangle$ shares the same optimal action as the original MDP $\langle \mathcal{S}, \mathcal{A}, R_{ext} \rangle$.*

**Theorem Conclusion:** *The above theorem shows that by adaptively adjusting intrinsic rewards, the optimal policy of the exploration-augmented MDP remains consistent with that of the original MDP. Thus, the introduction of additional rewards does not deviate the policy from the optimal direction, ensuring algorithmic performance. The detailed proof is provided in Appendix H.*

## 3 EXPERIMENTAL EVALUATION

### 3.1 EXPERIMENT SETTING

**Datasets.** To compare with previous work, we use the same prompt set consisting of 45 different animal classes for training (Simple-animals), which has been widely adopted for reward fine-tuning (Black et al., 2023; Yang et al., 2024a; Zhang et al., 2024). For testing, we generate 8 different noises for each animal class, totaling 360 test samples. For generalization evaluation, larger-scale prompt sets are further considered in testing as done in (Black et al., 2023), including 398 animal classes from ImageNet (ImageNet-animals) and the above 45 animals each concatenated with an activity (Activity-animals). *In addition, to verify that our method remains effective on larger and more complex datasets, we further report experimental results on Pick-a-Pic (Fig. 14), HPSv2*

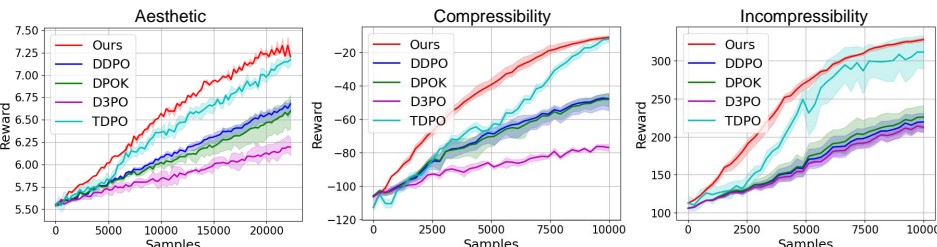

Figure 2: Reward curves of RL fine-tuning SDv14 with three distinct reward functions.

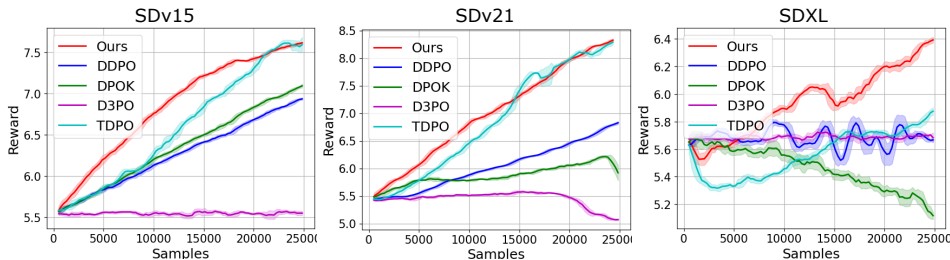

Figure 3: Reward curves of RL fine-tuning SDv15, SDv21, and SDXL on *aesthetic score*.

*(Fig. 8, Fig. 10), GenEval (Fig. 13), and HPS (Fig. 13).*

**Rewards and Metrics.** In this paper, we first deploy three metrics as the reward functions, that is, Aesthetic Score (Schuhmann et al., 2022), Compressibility, and Incompressibility. To quantitatively evaluate reward hacking, we then introduce cross-reward generalization and well-acknowledged metrics as suggested in (Zhang et al., 2024): ImageReward (IR) (Xu et al., 2023), Inception Score (IS) (Salimans et al., 2016), PickScore (PS) (Kirstain et al., 2023), and ClipScore (Clip) (Radford et al.). LPIPS (Zhang et al., 2018) is also introduced as a metric for evaluating diversity.

**SD Models.** To ensure fair comparison with existing baselines (Black et al., 2023; Yang et al., 2024a), we take Stable Diffusion v1.4 (SDv14) as our main test bed for reward fine-tuning. We also consider Stable Diffusion v1.5 (SDv15), v2.1-turbo (SDv21), and XL1.0 (XL) to verify the robustness of our method with different backbones.

*From 2D to 3D Diffusion Tasks. In addition to conducting experimental evaluations on 2D diffusion models for image generation tasks, we also perform both qualitative and quantitative experiments on a de-lighting task using 3D diffusion models. The results demonstrate that IntDiff is not only effective for 2D tasks but also achieves stable and significant performance improvements on 3D tasks (Fig. 23、Tab. 8).*

## 3.2 EXPERIMENTS ON PRE-DEFINED RL OBJECTIVES

The primary criterion for evaluating an RL fine-tuning algorithm is its ability to improve the target reward score. To this end, we compare our method against four state-of-the-art (SOTA) baselines (*i.e.*, DDPO (Black et al., 2023), DPOK (Fan et al., 2024), D3PO (Yang et al., 2024a), and TDPO (Zhang et al., 2024)) and employ three distinct reward functions as well as three different diffusion backbones to comprehensively assess the learning efficiency of our approach. As shown in Fig. 2, across all three reward functions applied to SD14, the learning curves consistently demonstrate that our method achieves more efficient RL training. Furthermore, Fig. 3 illustrates that our approach also leads to faster reward learning when applied to SD15, SD21, and SDXL. Specifically, on SD14 and SD15, Ours exhibits improved performance, especially at the beginning stage. Moreover, since SDXL is a more challenging model to be fine-tuned, only our method achieves obvious improvement toward the target reward. It further validates its superior performance across a broad range of scenarios. In Appendix D.1, we provide results trained on a larger-scale Pick-a-Pic dataset and further diversity and fidelity experiments.

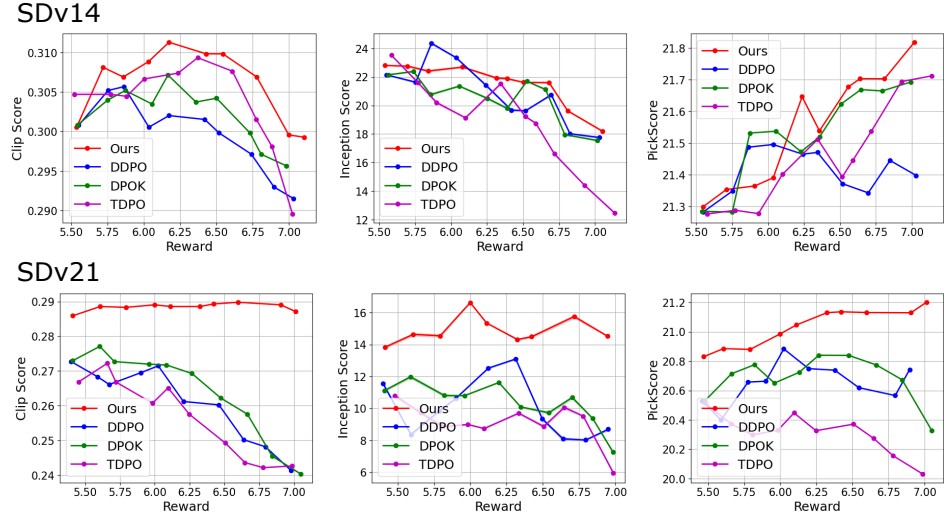

Figure 4: Image quality assessment at different levels of target rewards with Clip Score, Inception Score, and PickScore on two diffusion backbones.

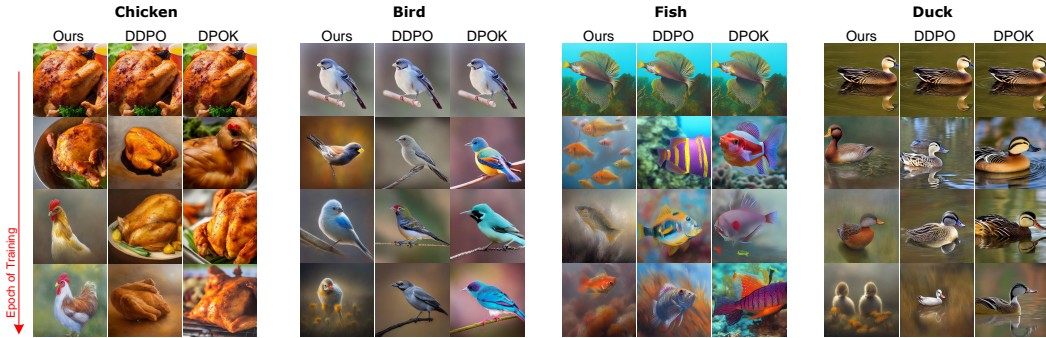

Figure 5: Image diversity across different training epochs. The seeds are the same fixed values for generating each group of images. Compared to Ours, the baselines perform limited in diversity from the perspective of *Semantic* (Group 1), *Object Posture* (Group 2&3&4), *Color* (Group 1&2&4), and *Object Count* (Group 3&4). In contrast, our method exhibits superior exploration ability during training, indicating its capability in finding improved optima to mitigate reward hacking. More results could be found in the Appendix.

## 3.3 DIVERSITY AND ALIGNMENT EVALUATION WITH MULTIPLE METRICS

Given the risk of reward hacking and overfitting to the specific target reward function, evaluating an algorithm solely based on the learning target reward is insufficient for comprehensive quality evaluations. Therefore, we introduce a suite of additional evaluation metrics that assess image quality from multiple perspectives, independent of the reward function used during training. These metrics include **PickScore (PS)** to measure human-perceived aesthetic quality, **CLIP Score (Clip)** to evaluate prompt-image alignment, and **Inception Score (IS)** to assess image diversity.

Since the learning efficiency with respect to aesthetic objectives can vary across methods, using the number of training samples as a control variable would yield unfair comparisons. Therefore, as shown in the Fig. 4, we adopt the aesthetic score (*i.e.*, the RL reward) as the control variable and plot each method's performance across various metrics at different levels of aesthetic score. The results demonstrate that our method consistently outperforms baselines in terms of diversity, human aesthetic preference, and prompt alignment, across a wide range of aesthetic scores.

Moreover, to further demonstrate that the incorporation of intrinsic rewards facilitates richer and more diverse exploration during training, we designed an experiment specifically aimed at evaluating

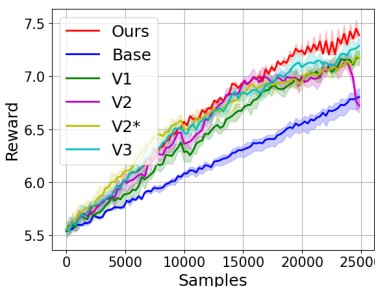

Figure 6: Ablation for each proposed component on Aesthetic Reward.

Table 1: Ablation study for each proposed component on cross-metric evaluation. All results are obtained from the models where Aesthetic Score=7.

| Var | Component | | | | Metric | | | | | |
|-----|-----------|---|---|---|--------|----|----|------|-------|------|
| | $R_t^{int}$ | $\Delta c_t$ | $\alpha_t$ | RDO | PS | IS | IR | Clip | Lpips | Top2 |
| Base | | | | | 21.46 | 17.51 | 0.766 | 0.295 | 0.615 | 0 |
| V1 | ✓ | | | | 21.69 | **18.67** | 0.931 | 0.292 | 0.627 | 1 |
| V2* | ✓ | ✓ | | | 21.71 | 18.42 | 0.951 | **0.308** | 0.628 | 1 |
| V3 | ✓ | ✓ | ✓ | | 21.75 | 18.39 | 0.964 | 0.302 | 0.636 | 3 |
| Ours | ✓ | ✓ | ✓ | ✓ | **21.80** | 18.51 | **1.027** | 0.302 | **0.638** | **5** |

exploration diversity. Rather than generating multiple images using different random seeds for a fixed-epoch model, we instead fix the random seed and generate images from models at different training epochs. This setup enables us to assess how a single input state evolves throughout training under the guidance of each algorithm, effectively capturing the trajectory of exploration. As shown in Fig. 5, with fixed seeds, the image sequences produced by DDPO and DPOK exhibit limited variation in terms of shape, color, and object count when compared to their pre-training outputs, suggesting that these methods engage in relatively conservative and constrained exploration. In contrast, our method exhibits continual exploratory behavior throughout training, yielding greater diversity and enabling the discovery of more effective trajectories for learning the reward function. Furthermore, thanks to the alignment-preserving regularization imposed by $\Delta c_t$, our method maintains strong alignment with the input prompts while simultaneously achieving higher generative diversity.

### 3.4 ABLATION STUDY

To validate the effectiveness of each proposed component, we conduct a series of ablation studies by designing multiple algorithmic variants. Specifically, we define the following configurations: 1) Base: the baseline method without any of the proposed components, corresponding to standard DDPO. 2) V1: Base augmented with intrinsic reward $R_t^{int}$. 3) V2: V1 with the addition of the $\Delta c_t$ regularization term. 4) V2*: V2 further extended with a fixed weight $\alpha_t$ for balancing the intrinsic reward. 5) V3: V2 with a dynamically adjusted weight $\alpha_t$ instead of a fixed one. 6) Ours: V3 with PPO replaced by RDO. For reward optimization, as shown in Fig. 6, each incremental addition of components leads to consistent performance improvements. Notably, applying the $\Delta c_t$ term without adjusting the intrinsic reward via $\alpha_t$ results in unstable training dynamics, potentially due to excessive exploration that causes the learning process to collapse at certain points—manifested as performance drops in the curve. In contrast, while using a fixed p provides more stability, it underperforms compared to the dynamic adjustment strategy.

In Tab. 1, we further report the performance of these variants on multiple unseen metrics, including those unrelated to the primary reward function. V2 is not shown in the table since it is proven unstable during training (see Fig. 6). The results collectively demonstrate the effectiveness and necessity of all proposed components in enhancing both reward optimization and generalization.

More ablation results about the details of $\Delta c_t$ and $\alpha_t$ could be found in the Appendix D.3.

### 3.5 RESULTS ON UNSEEN PROMPTS

**Cross-Dataset Animal Prompts.** Here, we evaluate the aesthetic score on ImageNet-animals and Activity-animals using the SDv14 model fine-tuned by prompts from Simple-animals. These datasets include more complex unseen prompts that are important for real-world applications. As shown in Fig. 7, our method achieves superior aesthetic scores on the unseen datasets, which indicates its cross-dataset effectiveness.

**Diverse&Complex Prompts.** In addition, we introduce more diverse and complex prompts for comprehensive evaluation. Specifically, we use the SDXL model fine-tuned by prompts from Simple-animals, and generate images based on a subset of prompts from the HPSv2 dataset (Wu et al., 2023a) (See *Supplementary Materials* for applied prompts). The subset comprises prompts of larger

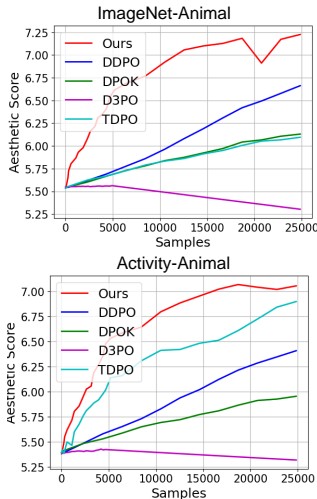
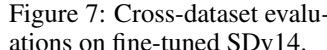

Figure 7: Cross-dataset evaluations on fine-tuned SDv14.

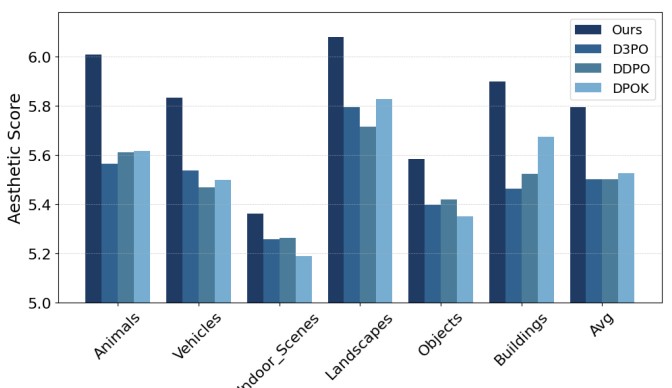

Figure 8: Diverse&Complex cross-prompt evaluations on fine-tuned SDXL. The results are generated by prompts from a subset of HPSv2 and grouped by categories. We also provide the average (Avg) aesthetic score in the last group.

complexity and broader semantic coverage beyond the animal domain. As shown in Fig. 8, we categorize the prompts into six classes and assess their aesthetic quality separately using the aesthetic score. The results across all classes consistently validate our superiority.

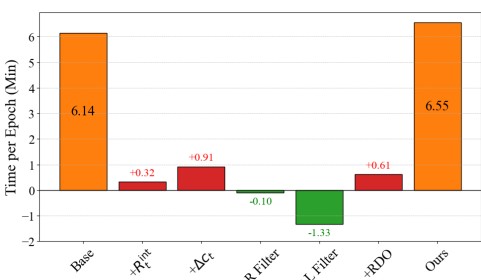

Figure 9: Contributions of each proposed component to training time consumption per epoch. Base refers to the basic pipeline, *i.e.*, DDPO.

Table 2: Overall Computation Efficiency of Baseline and Ours. The left part and right part are the metrics of overall training and a single batch of denoising, respectively.

| Method | Overall | | Single batch | |
|---|---|---|---|---|
| | Epoch | Time | GFLOPS | Time |
| DDPO | 117 | 11.98h | 1.034e+06 | 9.56s |
| DPOK | 128 | 14.76h | 1.034e+06 | 9.69s |
| **Ours** | 59 | 6.42h | 1.040e+06 | 5.96s |

### 3.6 COMPUTATION EFFICIENCY

Given that the proposed components in our method may affect both training and inference efficiency to varying degrees, we provide a detailed analysis of computational cost. As shown in Fig. 9, we first report the total time per training epoch, along with detailed time consumptions of each individual component contributed. While introducing intrinsic reward, the $\Delta c_t$ regularization term, and RDO introduces some additional overhead, the incorporation of Reward Filter and Length Filter effectively offsets this cost. As a result, the overall increase in per-epoch training time compared to the baseline is limited to only 6.7%. Moreover, given that our method significantly accelerates reward learning, the overall training efficiency is improved. This is further supported by the results in Tab. 2, where we compare the total training time required to reach an aesthetic score of 7 across different methods, as well as the computational cost of single-step denoising operations. Clearly, our approach demonstrates superior efficiency in both training duration and per-step inference cost.

### 3.7 SUBJECTIVE EVALUATION

To evaluate the subjective quality of the generated images, we conducted two complementary experiments: **Human Impression Evaluation** and **Scoring by Large Vision Model**. Specifically,

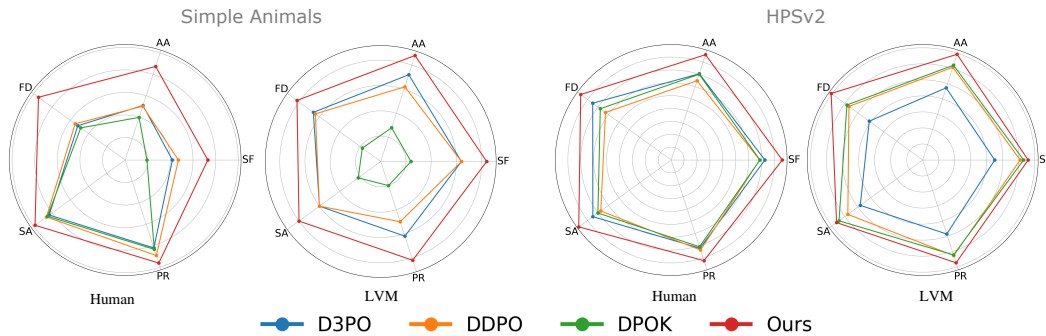

Figure 10: Subjective preference evaluation by human and Lagre Vision Model (LVM) on two prompt sets with five metric dimensions.

we employed the SD-XL model to generate images using both the simple animal and HPSv2 prompts across various methods. We designed five evaluation metrics—*Structural Faithfulness (SF)*, *Aesthetic Appeal (AA)*, *Fine-grained Detail (FD)*, *Semantic Alignment (SA)*, and *Prompt Responsiveness (PR)*—which were independently assessed by human subjects and by ChatGPT. Further experimental details of the user study are provided in the Appendix C.3. As the results shown in Fig. 10, our proposed IntDiff demonstrates superior performance across multiple dimensions in both human and large vision model evaluations.

## 4 CONCLUSION

In this paper, we propose a reinforcement learning (RL) framework tailored for fine-tuning diffusion models, systematically addressing key challenges including sparse rewards, reward hacking, and the exploration-exploitation trade-off. To this end, the proposed method incorporates intrinsic rewards within intermediate steps, introduces a semantically guided exploration filtering mechanism, and designs a denoising-progress-driven adaptive regulation strategy. These components collectively enhance both the diversity and semantic alignment of generated outputs while maintaining training stability and computational efficiency. Furthermore, a more efficient policy optimization approach is employed to improve sample efficiency and policy convergence quality. Extensive experiments demonstrate that the proposed framework achieves more effective optimization of predefined downstream objectives, while mitigating reward hacking and reducing computational overhead.

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

APPENDIX

# A   ADDITIONAL RELATED WORKS

## A.1   RL FOR DIFFUSION MODELS

Traditional likelihood-based training objectives can lead to a decline in image quality, as likelihood values often fail to reflect visual quality (Black et al., 2023) accurately. Reinforcement learning (RL) methods build on pre-trained text-to-image models, further fine-tuning them to directly optimize specific downstream task objectives, making generated images better aligned with task requirements. Among these, reward model-based methods, such as DDPO (Black et al., 2023) and DPOK (Fan et al., 2024), optimize generation quality by constructing reward functions to evaluate model outputs. On the other hand, human feedback-based methods, such as D3PO (Yang et al., 2024a), Diffusion-DPO (Wallace et al., 2024), and HPS (Wu et al., 2023b), leverage human preference scores directly, avoiding the complexity of reward modeling and aiming to align generation more closely with human aesthetic preferences. However, all these approaches face issues with delayed and sparse rewards or feedback (Franceschelli & Musolesi, 2024) and the balance between exploration and exploitation (Jena et al., 2024).

## A.2   REWARD HACKING

While reinforcement learning (RL) endows diffusion models with the capacity to align with predefined downstream objectives, it also introduces the risk of *reward hacking*—a phenomenon wherein the model, in the process of optimizing specific reward signals, progressively deviates from the original semantic intent and the true data distribution (Skalse et al., 2022; Gao et al., 2023; Zhang et al., 2024; Miao et al., 2025). This manifests as generations that excessively cater to the biases of the reward model, ultimately resulting in degraded semantic consistency and overall generation quality.

A prevailing class of approaches to counter reward hacking involves regularizing the deviation of the generation policy from the pretrained distribution. A representative method in this category is DPOK (Fan et al., 2024), which mitigates policy drift during fine-tuning by incorporating a Kullback–Leibler (KL) divergence regularization term. While such methods are effective to a certain extent in constraining policy shifts, their influence is predominantly limited to the distributional level, lacking fine-grained regulation over the semantic consistency evolution throughout the generation process. Consequently, they struggle to curb the gradual semantic drift effectively. Moreover, although KL divergence serves as a conservative constraint that stabilizes the generation policy, it also restricts extensive policy exploration when there is a significant mismatch between the reward signal and the original data distribution. This tension between reward alignment and distribution preservation can result in a narrowed policy space, thereby hampering the optimization of downstream objectives.

In contrast, the present work departs from the paradigm of indirectly constraining policy distributions. Instead, it intervenes directly within the generation process by constructing an explicit semantic deviation awareness mechanism and introducing structured exploration signals. This approach not only suppresses semantic drift but also facilitates effective expansion of the policy space, thereby enabling superior optimization of downstream objectives under the constraint of reward hacking prevention.

## A.3   INTRINSIC MOTIVATION

Intrinsic rewards (Pathak et al., 2017; Burda et al., 2018; Yan et al., 2024; 2023), as an effective mechanism to alleviate the problem of sparse external rewards, have been widely employed in reinforcement learning to encourage agents to explore intermediate states. However, existing intrinsic reward schemes, primarily designed for trial-and-error decision-making processes in conventional reinforcement learning, are not directly transferable to the diffusion generation paradigm. Moreover, the introduction of such mechanisms may further exacerbate the inherent exploration–exploitation dilemma characteristic of the reinforcement learning framework.

In this work, we propose a novel paradigm of intrinsic motivation tailored to the denoising process, leveraging the intermediate variables intrinsic to diffusion models. To the best of our knowledge,

this is the first attempt to incorporate intrinsic reward mechanisms into the fine-tuning of diffusion models. Specifically, we introduce an intrinsic incentive method suitable for the generative process and develop a corresponding adaptive coordination mechanism for exploration and exploitation. This mechanism dynamically adjusts the balance between exploration and exploitation based on intermediate feedback, thereby providing stable guidance to the generation policy. As a result, the proposed approach enhances training stability and improves performance on downstream tasks.

### A.4 *Intrinsic and Extrinsic rewards*

**Extrinsic Reward.** This reward serves as a task-level evaluation signal and is computed only at the end of the denoising process, i.e., on the final latent state. Although this form of reward is commonly adopted in most existing RL methods, its feedback is provided only at the terminal step, making it inherently sparse.

**intrinsic Reward.** The intrinsic reward is a task-specific exploration signal introduced in this work. It provides training guidance at all intermediate denoising steps, except for the final latent after denoising is complete. By delivering continuous feedback along the entire generation trajectory, the intrinsic reward effectively mitigates the signal deficiency that arises when relying solely on the terminal extrinsic reward. This prevents instability in policy learning caused by sparse end-point feedback.

### A.5 DIFFERENCES FROM EXISTING METHODS

#### A.5.1 THE DIFFERENCE BETWEEN INTDIFF'S INTRINSIC STEPWISE REWARDS AND CURIOSITY/NOVELTY BONUSES.

- A: IntDiff's intrinsic reward
- B: Curiosity/Novelty Bonuses

- *(1). Differences in State Reversibility (A. Reversible States vs B. Irreversible States)*: In RL tasks such as maze exploration, curiosity reward/novelty bonuses are based on the assumption that states are traversable and can be revisited, with novelty measured based on visit frequency or prediction error. In contrast, the state chain in a diffusion model is one-way and irreversible, and the same state cannot be revisited. Therefore, IntDiff's intrinsic rewards must rely on semantic progress, which is not a reuse or transformation of RL curiosity.

- *(2). Differences in Reward Paradigm (A. Single-State Rewards vs B. Cross-State Differential Rewards)*: In RL tasks like maze exploration, curiosity rewards/novelty bonuses are given based on a state that can be repeatedly reached. A classic approach is the "state-counting method" or "network-error method." In the state-counting method, novelty bonuses are typically designed as the inverse of the state visit frequency (e.g., $s$ represents the visit count of a single state $s_t$, and curiosity reward/novelty bonuses can be simply designed as $R_{\text{curiosity reward/novelty bonuses}} = 1/N + a$, where $a$ is a constant to avoid division by zero). The network-error-based curiosity rewards/novelty bonuses are defined as the error between the target network and predictor network outputs for a state $s_t$:

$$R_{\text{curiosity reward/novelty bonuses}} = \|f_{\text{target}}(s_t) - f_{\text{predictor}}(s_t)\|_2^2. \tag{8}$$

This error-based curiosity reward/novelty bonuses essentially approximate state-counting in a continuous manner, characterizing the visit density of a specific state. In difficult exploration tasks, curiosity rewards/novelty bonuses only involve the current state $s_t$. In contrast, the reward paradigm of the proposed diffusion model involves comparing the error between the current step $x_{t-1}$ and the previous step $x_t$ in the denoising process, involving two distinct and irrecoverable states.

- *(3). Differences in Exploration Behavior Constraints (A. Unconstrained Exploration vs B. High-Quality Exploration Under Semantic Supervision)*: RL curiosity rewards promote unconstrained exploration, rewarding any "new" state, regardless of direction. Our intrinsic reward system encourages directional exploration under semantic supervision: - $R_t^{int}$ is responsible for density enhancement, - $\Delta$CLIP determines semantic direction, penalizing

deviations. In other words, RL curiosity rewards promote "new state exploration," while our approach retains novelty only if it aligns with semantic constraints.

- *(4). Differences in Exploration Directionality (A. Directionless Exploration vs B. Directional Exploration) (This is the most distinguishing feature)*: RL curiosity promotes directionless, random exploration—rewards are given for any new state. In contrast, the intrinsic reward mechanism in our diffusion model, constrained by $\Delta$CLIP, ensures strong directional exploration guided by semantics. Only perturbations in the correct semantic direction are rewarded, and deviations are immediately penalized. Curiosity rewards are "random walks," while IntDiff follows "targeted semantic walks."

- *(5). Differences in Reward Focus (A. State Visit Count vs B. Reasonable Perturbation of Distribution)*: Curiosity rewards focus on "how many times a state has been visited," which is a function of visit frequency. IntDiff rewards focus on "whether the perturbation of the current image distribution is moving in the correct semantic direction," which is a function of distribution evolution quality. The former rewards "exploring new places," while the latter rewards "seeking new compositional strategies while approaching the target distribution." The former is about finding new paths, while the latter is about finding new compositional strategies.

- *(6). Differences in the Use of Auxiliary Modules (A. Introduction of Auxiliary Networks/Modules vs B. No Auxiliary Modules Needed)*: The components in curiosity rewards typically rely on additional networks such as prediction networks, inverse dynamics networks, or visit counters. In contrast, the intrinsic rewards of our diffusion model do not require any auxiliary modules. All necessary variables come from the latent differences between two adjacent steps in the denoising process of the diffusion model.

- *(7). Differences in Reward Function (A. Penalizing Excessive Single-State Visits vs B. Encouraging Compositional Manifold Changes)*: Curiosity rewards function to discourage repeated visits to the same state, essentially telling the agent not to "keep spinning around the same point." In contrast, IntDiff's reward function regulates the local shape, details, and contours of an image in the process from $t$ to $t-1$. The former controls "repetition" based on visit counts, while the latter regulates "shape transformation at the distribution level."

- *(8). Differences Between Unidirectional Reward Mechanisms and Bidirectional Control Rewards (A. Traditional Exploration Rewards (Only Positive) vs B. Bidirectional Reward Signals (Positive and Negative))*: IntDiff's intrinsic rewards automatically become negative when semantic consistency decreases, thus preventing jumps that deviate from the semantics. In contrast, traditional curiosity/novelty rewards are always positive by design, providing only a one-way drive of "the more exploration, the better." This structural difference allows IntDiff to achieve "exploration quality control" rather than simply increasing the "amount of exploration."

In summary, the intrinsic rewards of our diffusion model are significantly different from existing curiosity reward/novelty bonuses in terms of composition, target, dimension, mechanism, and function. They are not similar or close in nature.

### A.5.2 The Difference Between IntDiff's Semantic Consistency Regularization and CLIP-gated Faithfulness.

- A. CLIP-gated Faithfulness

- B. IntDiff's Semantic Consistency Regularization

- *(1). Distinguishing Positive and Negative Rewards (A. Unidirectional Filtering vs B. Bidirectional Control)*: Semantic Consistency Regularization (SCR) can provide positive or negative rewards for the same action depending on the sign of $\Delta$CLIP. In contrast, CLIP-gated faithfulness performs a unidirectional filtering process:

  Passed $\rightarrow$ Accepted

  Failed $\rightarrow$ Discarded

  SCR has bidirectional gradient signals, whereas CLIP-gated faithfulness only applies a unidirectional constraint signal. This means SCR is essentially a differentiable reward shaping mechanism, while CLIP-gated faithfulness is merely a gating mechanism.

- *(2). Constraints and Optimization in Parallel (A. Simple Gating vs B. Collaborative Optimization)*: CLIP-gated faithfulness is solely responsible for filtering specific samples and is not optimized itself—it serves as a "static threshold." In contrast, our SCR uses $\Delta$ CLIP to control the reward's sign, both constraining exploration and guiding the direction. As shown in Eq. 5 the basic intrinsic reward term is always positive, so the final reward's sign is entirely determined by $\Delta$CLIP. This drives the policy toward an exploration direction that increases CLIP alignment. As the policy optimizes, the CLIP alignment is indirectly improved as well. This represents a "constraining others + self-improvement through feedback" collaborative optimization mechanism.

In summary, SCR provides directional guidance for exploration, while the exploration behavior in turn enhances the CLIP metric within SCR. Fig. 6 demonstrates that CLIP remains unbiasingly aligned throughout training, which is the key reason why our method consistently outperforms others.

### A.5.3 THE DIFFERENCE BETWEEN INTDIFF'S ADAPTIVE AND EARLY-EXIT SCHEDULERS.

- A. Early-Exit Schedulers:Only capable of early stopping (single early stopping function).

- B. IntDiff's Adaptive: Adaptive control of denoising process exploration intensity + adaptive early stopping (a mechanism with dual benefits: balancing exploration and exploitation through adaptive collaboration + early stopping).

- (1). The main difference lies in our adaptive mechanism, which not only serves as a criterion for early stopping but, more importantly, dynamically balances the exploration and exploitation weights at different stages of denoising during training. In the early stages, when the agent is far from the final sparse reward, it strengthens exploration. As the denoising progresses and the generated output approaches the final sparse reward, the training focus gradually shifts to maximizing the preset task reward (exploitation).

- (2). Exploration-exploitation trade-off has long been an unresolved issue in RL, and methods for adaptively balancing these two are few. When RL is applied to fine-tune diffusion models, the exploration-exploitation dilemma is exacerbated by sparse rewards and long sequence decision-making. Our adaptive mechanism offers a solution to this issue.

Summary: Our adaptive mechanism not only allows for early stopping but also dynamically adjusts the exploration intensity, enabling an adaptive balance between exploration and exploitation.

### A.5.4 THIS FRAMEWORK IS A TAILORED TRAJECTORY-LEVEL ADAPTIVE INTELLIGENT DECISION SYSTEM FOR DIFFUSION MODELS, NOT A REASSEMBLY OF EXISTING METHODS.

- *(1). Bidirectional Collaborative Optimization*: Semantic Consistency Regularization and IntDiff's intrinsic stepwise rewards are not a simple combination; this combination realizes a bidirectional collaborative optimization.

  Existing Description: We have already specifically described collaborative optimization in lines 216-226 of the main text.

  Further Details: Our method forms a bidirectional collaborative feedback loop between semantic consistency regularization (SCR) and exploration behavior. SCR controls the reward direction through the sign of $\Delta$CLIP, thereby limiting exploration deviations from semantics. On the other hand, the policy updates tend to choose actions that improve semantic consistency, leading to continuous reverse optimization of the SCR's semantic alignment target during training.

  Ultimately, a stable positive feedback loop is formed: Semantic constraint exploration $\rightarrow$ better exploration results $\rightarrow$ further improvement of semantics.

  In summary, first, the IntDiff's intrinsic stepwise rewards and Semantic Consistency Regularization are independently distinguished from the methods you mentioned. Under these differing premises, the combination of Semantic Consistency Regularization and IntDiff's intrinsic stepwise rewards is not a simple combination but a design intended to achieve collaborative optimization.

- *(2). Adaptive Collaboration Between Exploration and Exploitation*:

After achieving the adaptive collaborative optimization between Semantic Consistency Regularization and IntDiff's intrinsic stepwise rewards, we further introduce an adaptive collaborative mechanism. This is based on the consideration of adaptive collaboration between exploration and exploitation, rather than simply combining mechanisms (please refer to (3) for further clarification).

# B  OPTIMIZATION DETAILS

## B.1  REFLECTIVE DIFFUSION OPTIMIZATION

After introducing the intrinsic reward (Section 2.2) and the step truncation mechanism (Section 2.4), improving sample utilization under sample filtering and inference compression becomes a key issue. Traditional policy optimization methods struggle to effectively leverage intermediate and future information in sequences, which limits policy learning efficiency. To address this, we introduce Reflective Diffusion Optimization (RDO), which possesses trajectory-reflective capability and is the first to apply such a method to diffusion model reinforcement learning fine-tuning.

RDO constructs an auxiliary optimization term by incorporating current and future state–action pairs, guiding the policy to use trajectory posteriors to improve update stability and sample efficiency. The optimization objective maintains the backbone structure of conventional RL fine-tuning frameworks, and adds a future trajectory-based term estimated using importance sampling:

$$\hat{L}(p, p_{old}) = \mathbb{E}_{(s,a)} \left\{ \sum_{t=0}^{T} p_\theta(\mathbf{x}_t|c, \mathbf{x}_{t-1}) A_t^{p_{\theta_{old}}} + \beta \sum_{t=0}^{T-1} p_\theta(\mathbf{x}_t|c, \mathbf{x}_{t-1}) p_\theta(\mathbf{x}_{t+1}|c, \mathbf{x}_t) A_{t+1}^{p_{\theta_{old}}} \right\} \quad (9)$$

where $A^{\hat{\pi}}$ is the advantage function under the old policy $p_{\theta_{old}}$, and $\beta$ is a hyperparameter. The expectation is taken over denoising trajectories generated by the parameters $\theta_{old}$.

This optimization strategy fully exploits the multi-step structure of diffusion generation, using future states to improve the quality of current policy updates. It alleviates instability caused by sample redundancy and update noise. Under enhanced exploration via intrinsic motivation and step reduction via truncation, RDO provides a stable and efficient optimization path that ensures sample efficiency and final generation quality. Experimental results demonstrate that this method outperforms conventional policy optimization baselines in improving target scores and image-text alignment (see Fig. 6 and Tab. 1).

# C  EXPERIMENTAL DETAILS

## C.1  HYPER-PARAMETERS

We train our model with Adam optimizer (Kingma, 2014) and a learning rate of $3 \times 10^{-4}$. The full list of hyper-parameters in our paper is shown in Table 3. For each method and each RL objective, we ran five different seeds and report the mean and standard deviation of reward on 64 randomly sampled prompts as validation set. We train each model with a total number of 25000 samples. Each experiment is conducted on a single machine with 8 NVIDIA A100 GPUs. Following (Black et al., 2023), we use the LAION aesthetics predictor for conducting the aesthetics experiments.

## C.2  IMPLEMENTATION DETAILS OF ADAPTIVE BALANCE PREDICTOR

In Subsection 2.4, we introduced a Adaptive Balance Predictor that learned to predict the dynamic coefficient of the intrinsic reward according to how close the image in the current de-noising step is to the fully denoised real images. The architecture of the perception model is a simple four-layer CNN followed by a fully connected layer and a Sigmoid layer for binary classification. The kernel sizes are (3, 3, 3, 1) and channels are (64, 128, 256, 256).

To train the adaptive balance predictor, we update its parameters after each optimization step of the RL loss, with data from the current de-noising chain. To construct the training data, given a $T$-step de-noising process, we take the first $n$ images near $\{\mathbf{x}_T\}$ as negative samples and the last $n$ images $\{\mathbf{x}_0\}$ as positive samples, where $n = 5$. In addition, we set to $\alpha_t$ to 1 for the last five de-noising steps near $\mathbf{x}_0$, which makes the intrinsic rewards to zero for almost real images.

| Name | Description | Value |
|---|---|---|
| $lr$ | learning rate of IntDiff | 3e-4 |
| optimizer | type of optimizer | Adam |
| $\xi$ | weight decay of optimizer | 1e-4 |
| $\epsilon$ | Gradient clip norm | 1.0 |
| $\beta_1$ | $\beta_1$ of Adam | 0.9 |
| $\beta_2$ | $\beta_2$ of Adam | 0.999 |
| $T$ | total timesteps of inference | 50 |
| $bs$ | train batch size per GPU | 2 |
| $bs_{sample}$ | sample batch size per GPU | 8 |
| $n$ | number of batch samples per epoch | 4 |
| $\eta$ | eta parameter for the DDIM sampler | 1.0 |
| $G$ | gradient accumulation steps | 4 |
| $w$ | classifier-free guidance weight | 5.0 |
| $mp$ | mixed precision | fp16 |

Table 3: Hyper-parameters in our experiment.

### C.3 HUMAN&LVM EVALUATION PROTOCOL

For a more accurate evaluation of the models after fine-tuning for alignment, we conducted a human&LVM evaluation (Fig. 10). Firstly, the applied simple animal and HPSv2 prompts along with the corresponding images generated by the four methods could be found in the *supplementary material*. Secondly, we provide the detailed definition of all five metrics that we also described to both human subjects and LVM:

- **Semantic Alignment**: Does the generated image accurately and comprehensively convey the key entities, scenes, and actions described in the text? (Measures the degree of direct correspondence between the textual content and the visual output.)

- **Structural Faithfulness**: Does the image exhibit any structural anomalies, such as disproportionate elements, extra limbs, or misaligned backgrounds? (Assesses the logical coherence and structural plausibility of the visual composition.)

- **Aesthetic Appeal**: Which image demonstrates superior appeal in terms of visual style, composition, color harmony, and overall aesthetics? (Reflects traditional notions of visual attractiveness.)

- **Fine-grained Detail**: Which image exhibits greater finesse and naturalism in rendering textures, materials, shadows, and other fine-grained visual details? (Reflects perceptual resolution and detail fidelity.)

- **Prompt Responsiveness**: Which image more accurately reflects the attribute constraints specified in the prompt, such as color, quantity, or action? (Used to evaluate the controllability and precision of prompt adherence.)

For Human Evaluation, we employed five subjects to evaluate all images from the five metrics in a method-blind manner. For LVM experiments, we leveraged the web version of the powerful GPT o4-mini-high by sending it all images with a request to evaluate the metrics.

## D    FURTHER EXPERIMENTS

### D.1    SUPPLEMENTING RESULTS FOR GENERATED IMAGE DIVERSITY AND ALIGNMENT

In Sec. 3.3, we have provided results with PickScore, Clip Score, and Inception Score on SDv14 and SDv21. Here, we supplement further results, including metric ImageReward and backbone SDv15. As shown in Fig. 11 and 12, the results consistently demonstrates the superiorty of our method.

Moreover, besides the simple-animal dataset, we further conduct experiments on the complex and fantastic Pick-a-pic dataset in Fig. 14. In Fig. 13, we additionally introduce *the well-established alignment dataset GenEval Ghosh et al. (2023), and the photorealistic dataset HPSv2 Wu et al.*

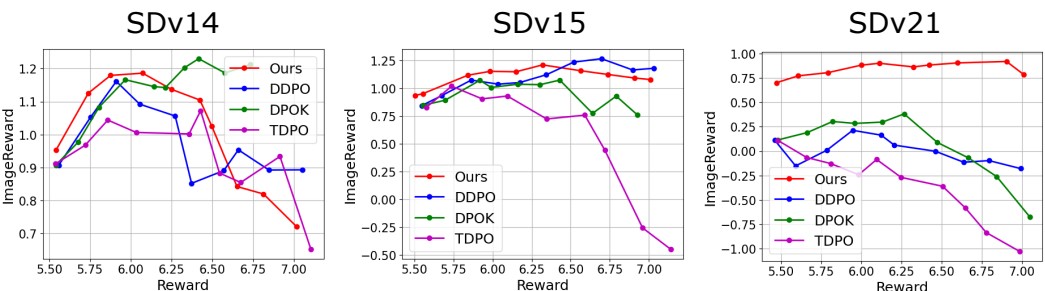

Figure 11: ImageReward evaluation on SDv14, SDv15, and SDv21.

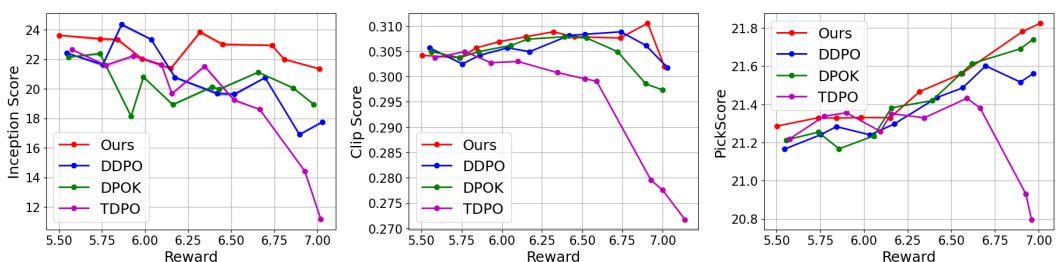

Figure 12: Clip Score, Inception Score, and PickScore on SDv15.

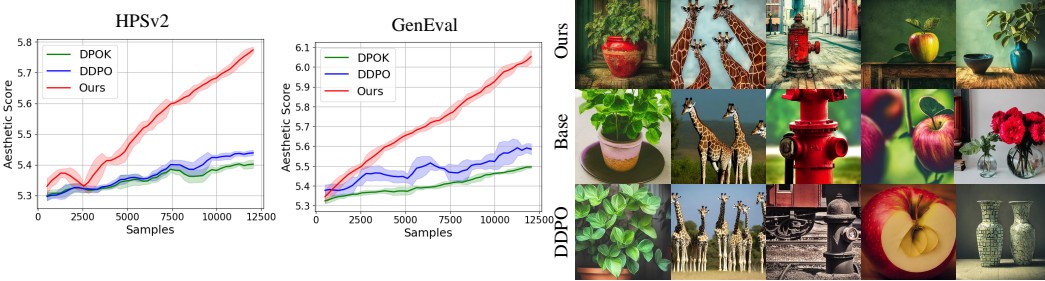

Figure 13: *Results on the complex GenEval and HPSv2. Our results achieve a more appealing aesthetic and alignment performance.*

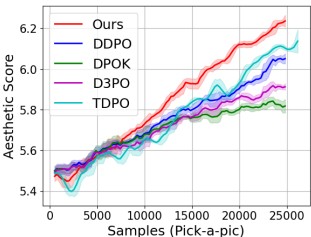

Figure 14: Results trained on Pick-a-pic dataset.

Table 4: Comparison of different methods across Diversity and Fidelity metrics.

| Method | AES | Diversity | | | Fidelity | | | #Top2 |
|--------|-----|-----------|-----|-----|----------|-----|------|-------|
| | | LPIPS↑ | TCE↑ | IS↑ | FID↓ | iFS↑ | CLIP↑ | |
| Diff-DPO* | 5.66 | 0.639 | 37.97 | 21.60 | 99.38 | 0.813 | 0.313 | - |
| DDPO | 7.03 | 0.625 | 38.04 | 18.01 | 139.9 | 0.764 | 0.312 | 2 |
| DPOK | 6.98 | 0.602 | 36.63 | 19.99 | 118.7 | 0.859 | 0.301 | 4 |
| TDPO | 7.01 | 0.563 | 31.55 | 11.93 | 180.3 | 0.371 | 0.279 | 0 |
| Ours | 7.01 | 0.641 | 38.26 | 21.56 | 91.62 | 0.883 | 0.313 | 6 |

*(2023a). The results consistently demonstrate that our method is capable of optimizing complex prompts with superiority. We provide the visual impression in comparison with DDPO, and the base model in Fig. 13 right.*

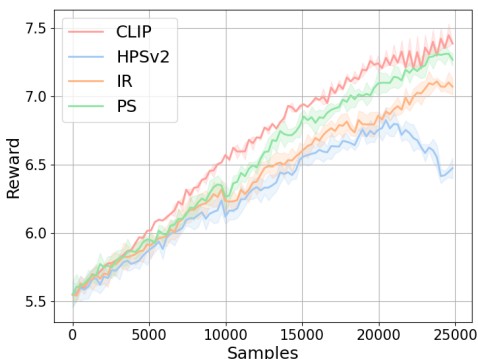

Figure 15: Effectiveness of different $\Delta c_t$ evaluators

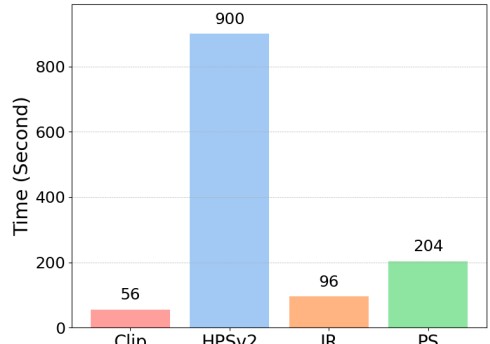

Figure 16: Efficiency of different $\Delta c_t$ evaluators

## D.2 COMPARISON ON GENERATED IMAGE DIVERSITY AND FIDELITY

The diversity and fidelity of generated images are highly related to the issue of reward hacking, since the hacked results usually exhibit a similar pattern or color. Here, we compare our method with DDPO, DPOK, TDPO, and Diff-DPO (gradient-based training) based on five metrics: FID and improved F1 Score (iFS) for Fiderlity, and IS, LPIPS, and TCE for Diversity. We use the official pretrained Diff-DPO model with AES=5.66 for evaluation, and all other methods are reproduced with AES reaches 7. D3PO is not considered for comparison due to its ineffective optimization on SDv15. The prompt set is still simple-animal with 8 seeds. FID and iFS are calculated by taking the generated images of the original SD backbone as the references. As shown in Tab. 4, DPOK has certain advantage on preserving Diversity and Fidelity, while TDPO entirely fails to maintain them. Nevertheless, our method achieves the best performance in both dimensions, which demonstrates its effectiveness in mitigating reward hacking.

*Moreover, we consider the inter-prompt diversity. As shown in Tab. 5, we report the mean LPIPS and Truncated CLIP Entropy (TCE) on each prompt in simple animal set with 32 seeds, SDv14, and AES=7. Our results clearly demonstrate the superiority of diversity by mitigating reward hacking.*

Table 5: *Inter-prompt diversity comparisons on simple animal set.*

| Metric | DDPO | DPOK | D3PO | TDPO | Ours |
|--------|------|------|------|------|------|
| LPIPS | 0.631 | 0.595 | 0.591 | 0.560 | 0.637 |
| TCE | 38.02 | 36.70 | 36.32 | 31.93 | 38.18 |

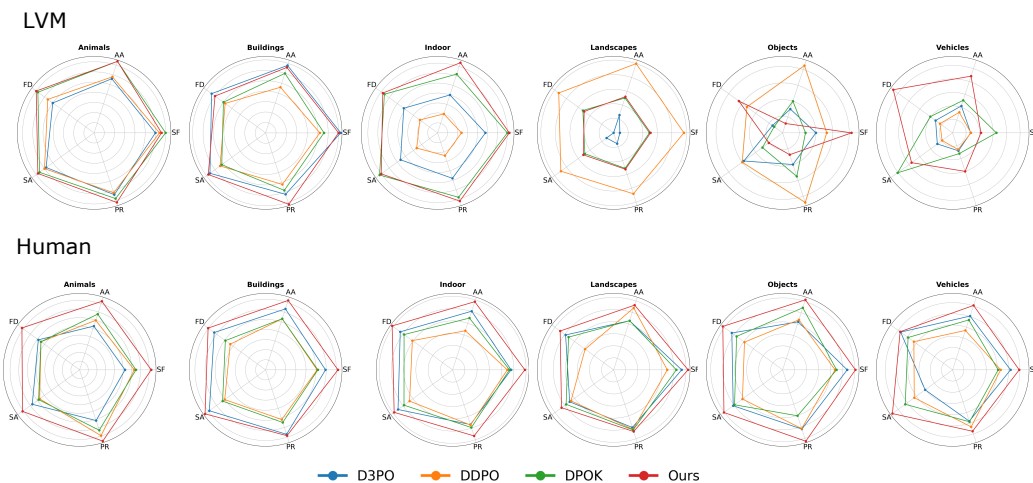

Figure 17: Detailed user study for human and LVM on HPSv2 prompt set.

## D.3 MORE ABLATION STUDY

**Effect of Adaptive Balance Predictor** As a critical component for assessing denoising performance under prompts of varying difficulty, the adaptive balance predictor plays a dual role: it participates in the dynamic adjustment of the intrinsic reward and also functions as a truncation discriminator. It is trained jointly with the fine-tuned model and gradually learns the denoising progression of the current fine-tuning process, continuously adapting as training proceeds. In Sec. 3.4, we have experimentally demonstrated the positive impact of dynamic $\alpha_t$ over both the w/o $\alpha_t$ and fixed $\alpha_t$ baselines in terms of effectiveness. Here, we first present the learning trajectory of $\alpha_t$ during training. As illustrated in Fig. 18, the results indicate that $\alpha_t$ transitions from initially 50 (applying no termination) to converging towards a stabilized mean value around 32. This outcome is consistent with our design expectations for the predictor.

**Effect of $\Delta c_t$ Evaluator.** In addition to CLIP, alternative evaluators such as HPSv2, IR, and PS can also be employed for $\Delta c_t$ calculation to judge the alignment and aesthetic metrics. Accordingly, we evaluate these alternatives from the perspectives of performance and computational cost. As illustrated in Fig. 15, CLIP and PS exhibit comparable RL effectiveness, whereas HPSv2 and IR perform slightly worse. In terms of computational efficiency, Fig. 16 clearly demonstrates that CLIP is the fastest among the evaluated methods. In summary, employing CLIP as the $\Delta c_t$ evaluator constitutes the most concise and effective choice.

**Quantitative Evaluation on Images with Length Filter.** We also investigate the generated images under with length filter. Under the condition where the mean AesScore reaches 7 with 50 denoising steps, we compared models trained with DDPO, DPOK, and our proposed approach. For our method, length filter is performed dynamically via the predictor, whereas for DDPO and DPOK, we adopted the average of the dynamic length filter values as the fixed length. As shown in Tab. 6, our method consistently outperforms the baselines across multiple metrics, including Aesthetic Score (Aes), CLIP Score (Clip), PickScore (PS), ImageReward (IR), and Inception Score (IS). These results indicate that our method accelerates the denoising process through high-quality exploration.

## D.4 DETAILED USER STUDY ON HPSV2 PROMPT SET

Moreover, considering the deployed HPSv2 prompt set includes six categories, in Fig. 17, we present the detailed evaluation results specific to each category. The results exhibit more diversity while our method could maintain superiority under overall consideration.

## D.5 DETAILS OF COMPUTATIONAL COST

*Here, we provide the detailed computational cost on SDv14 to achieve AES=7.0. As shown in Tab. 7, all results are obtained from NVIDIA official command and W&B record. It can be observed that*

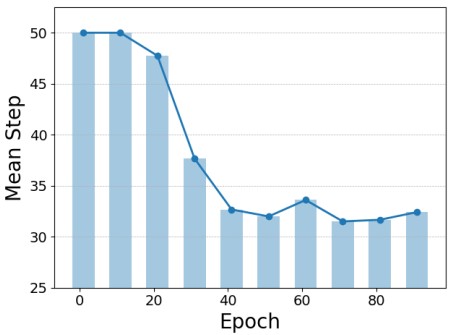

Figure 18: Mean denoising step of our method during training

Table 6: Quality assessment of images generated by fewer steps. Ours are generated by adaptive step, which is predicted and calculated by $\alpha_t$ predictor. DDPO and DPOK are generated by a fixed step of 32, which is the average step of Ours.

| Method | Aes | Clip | IR | IS | PS |
|---|---|---|---|---|---|
| DDPO | 0.6675 | 0.2819 | 0.7126 | 16.27 | 20.98 |
| DPOK | 0.6537 | 0.2931 | **0.7545** | 16.15 | 21.40 |
| **Ours** | **0.6810** | **0.2957** | 0.6971 | **17.94** | **21.69** |

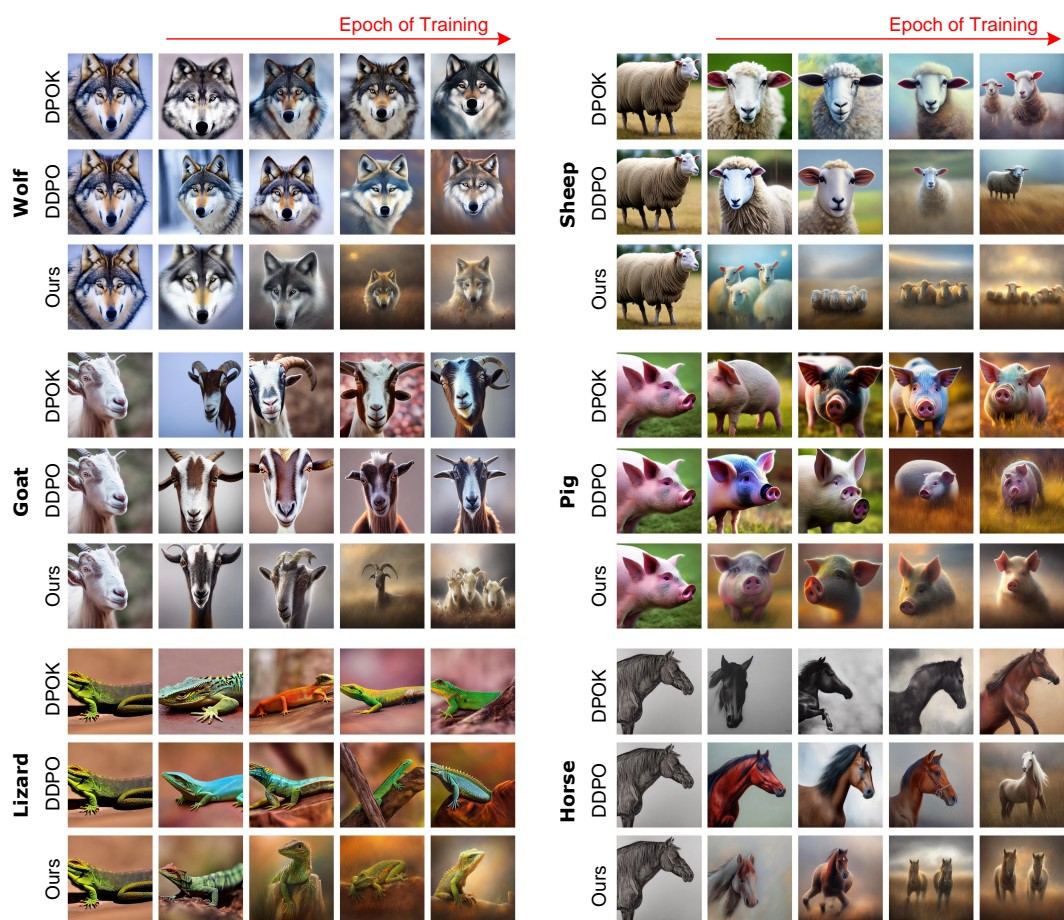

Figure 19: Further visual results of images generated with fixed seeds from different epochs.

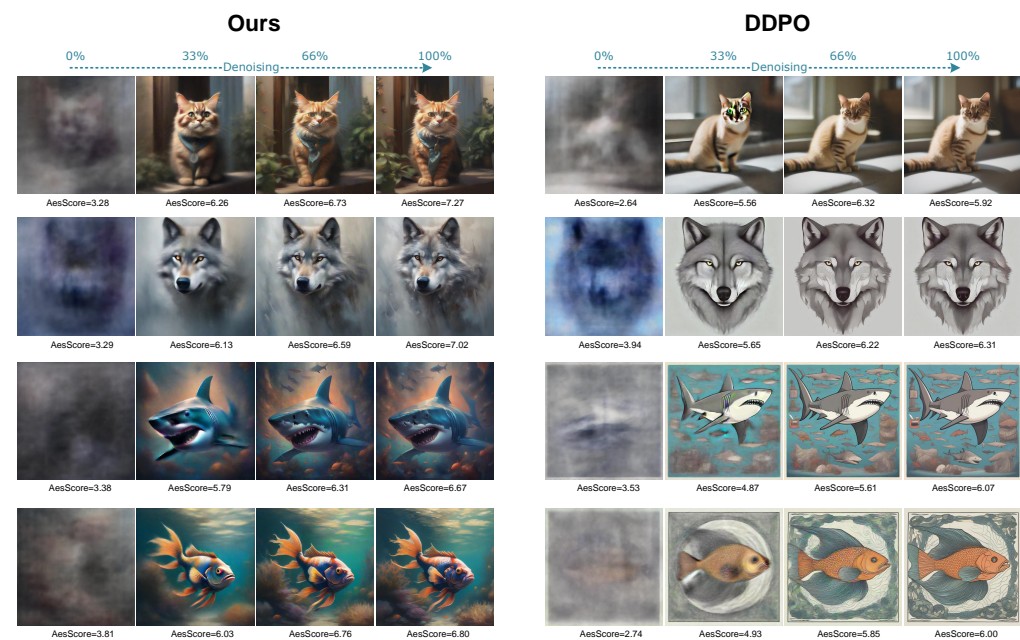

Figure 20: Visual results of early terminated images. The results are decoded from the intermediate latent codes during denoising.

Table 7: *Training and inference computational cost. The results are obtained with 8 A100 GPUs when training, and single H100 GPU when inference. WH, GH, PE, GM refer to wall-clock hours, GPU-hours, peak energy (W), and GPU memory per GPU (GB), respectively. We train with batch size of 32 and inference with batch size of 12.*

| Method | Training | | | | Inference | | | |
|--------|------|-------|---------|-------|-------|-------|--------|-------|
| | WH | GH | PE | GM | WH | GH | PE | GM |
| DDPO | 11.98 | 88.40 | 1994.88 | 67.21 | 0.006 | 0.005 | 348.25 | 20.97 |
| Ours | 6.42 | 48.08 | 2207.76 | 76.13 | 0.006 | 0.005 | 348.17 | 20.95 |

*our method effectively accelerates the training process, with acceptable memory and energy cost for 80GB GPUs. During inference, our method is the same as the baseline DDPO>*

### D.6 MORE VISUAL RESULTS

**Exploration across Training** We supplement more visual results of the image sequence generated by the fixed seed on different epochs in Fig. 19. Compared to DDPO and DPOK, our method exhibits richer diversity during training.

**Early Terminated Image** In Fig. 20, we showcase the images that are denoised with early termination. The results indicate two standpoints: 1) The coarse-grained structures of images are determined in the early stage of denoising. Therefore, the early termination will not severely disrupt the final generation quality of images. 2) Compared to DDPO, our results exhibit superior aesthetic performance across all denoising stages consistently.

*Visual Impression of Reward Hacking As shown in Fig. 21, we present visual results for a direct impression of the effect of reward hacking. It can be observed that with the same levels of high AES score, DDPO and TDPO fall to different local optima that are both reward hacking: DDPO tends to be all red with lots of grass, TDPO tends to have golden shining backgrounds and tiny figures. All of these generated images have poor fidelity and diversity, that is, suffering from severe reward hacking.*

DDPO *(AES>7)*      TDPO *(AES>7)*      Ours *(AES>7)*

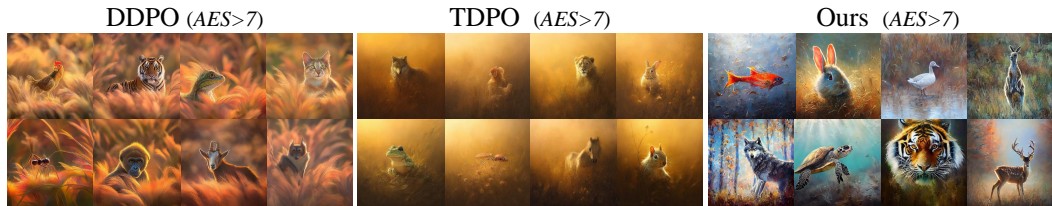

Figure 21: *Illustration of Reward Hacking. The baselines suffer from the lack of diversity and fidelity when aesthetic scores are high, while our results still maintain them, thanks to the directional exploration to find a better local optimum.*

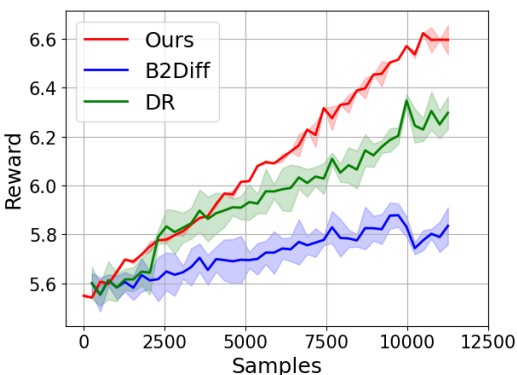

Figure 22: *More comparing results on SDv14 with simple animal and aesthetic score.*

## D.7 COMPARING TO MORE ADVANCED SOTA

*Here, we further compare the performance with more advanced SoTa methods, including DenseReward Yang et al. (2024b) and B2Diif Hu et al. (2025). As shown in Fig. 22, our method still consistently surpasses the existing SoTA methods.*

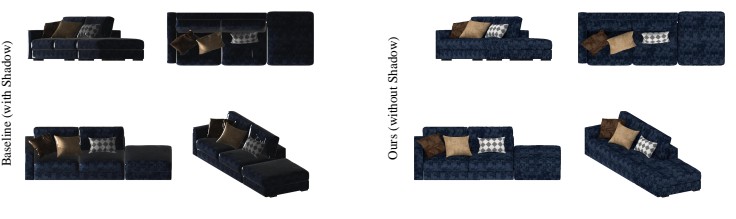

Figure 23: *3D objects with (baseline) or without (Ours) shadows.*

Table 8: *Implementing our method on 3D texture mapping models, with de-Shadow as reward.*

| Method | Shadow$^{\downarrow}$ | PS | CLIP |
|---|---|---|---|
| Base | 0.27/15.38% | 21.87 | 0.240 |
| Ours | 0.04/3.2% | 22.09 | 0.241 |

## D.8 RESULTS TRANSFERED TO 3D MODELS.

*Here, we investigate the integration of IntDiff into 3D generation tasks. Specifically, we employ a shading-aware scorer as a reward function to encourage shadow removal from 3D models. As demonstrated in Tab. 8 and Fig. 23, our method proves to be effective when applied to 3D-related tasks.*

## E ALGORITHM

We provide the overall algorithm of the proposed IntDiff in Alg. 1.

---

**Algorithm 1:** Algorithm of IntDiff

---

**Input:** Prompt set: $\mathcal{P}$; Training epoch: $\mathcal{E}$; Denoising step: $\mathcal{T}$.

1   Initialize pretrained diffusion model $\epsilon_\theta$;

2   **for** $e = 1$ **to** $\mathcal{E}$ **do**

3     **for** *Prompt $p$ in $\mathcal{P}$* **do**

4        generate sample trajectory of $p$ iteratively

5        $\{\mathbf{x}_{\mathcal{T}-1}^p, ..., \mathbf{x}_0^p\} = \{\mu(\mathbf{x}_{\mathcal{T}}^p, t) + \sigma_{\mathcal{T}}\mathbf{I}, ..., \mu(\mathbf{x}_1^p, 1) + \sigma_1\mathbf{I}\}$

6        compute extrinsic reward

7        $R_{\text{ext}} = r(\mathbf{x}_0^p, \mathbf{c})$

8        **for** *Timestep $t$ in reversed $\mathcal{T}$* **do**

9           perform one step of denoising

10          $\mathbf{x}_{t-1}^p = \mu(\mathbf{x}_t^p, t) + \sigma_t\mathbf{I}$

11           predicting $\alpha$

12          $\alpha_{t-1} = f(\mathbf{x}_{t-1}^p)$

13          **if** $\alpha_{t-1} \approx 1$ **do** break

14           compute intrinsic reward

15          $R_{\text{int}}^{(t-1)} = \left\| \hat{\mathbf{x}}_0^{(t)} - \hat{\mathbf{x}}_0^{(t-1)} \right\|_2^2$

16           compute $\Delta c_t$ with CLIP as $F$

17          $\Delta c_t = F(\hat{\mathbf{x}}_0^{(t)}, \mathbf{y}) - F(\hat{\mathbf{x}}_0^{(t-1)}, \mathbf{y})$

18           Filtering negative $\Delta c_t$

19          $\Delta c_t^* = Filter(\Delta c_t)$

20           optimizing both rewards via $\hat{L}(p_\theta, p_{\theta_{old}})$ in Eq.(9)

**Output:** learned model parameter $\theta$.

---

## F    LIMITATION AND ETHICAL IMPACT

**Limitation.** The proposed IntDiff could significantly mitigate the reward hacking issue, resulting in superior generated image quality and diversity. However, we believe that the reward hacking issue may be inevitable with the infinite extension of the training process. Therefore, it is challenging to determine at which specific point IntDiff achieves reward maximization while remaining largely unaffected by the adverse impact of reward hacking. In this study, we approximate such a scenario by conducting extensive experiments under the setting of reward = 7. However, this does not necessarily correspond to the most optimal value of the reward. Identifying this optimal value remains an open question for future investigation.

**Ethical Impact.** The proposed IntDiff enhances the applicability of Diffusion Models to downstream tasks, thereby improving both production quality and efficiency. Although the potential for malicious use of generated data cannot be entirely ruled out, the high-quality generation results presented herein offer more advanced samples for training improved generation detection modules, thereby facilitating enhanced performance in generation detection for privacy and security.

## G    DECLARATION OF LLM USAGES.

An LLM was used in part to refine the language for better presentation. The conceptualization, theoretical derivations, and experimental procedures were performed entirely by the authors without LLM assistance.

## H    PROOF OF THEOREM 2.1

*Proof.* According to the Bellman function of RL, defined the state-action value function with intrinsic rewards $\hat{Q}_t^\pi \triangleq \mathbb{E}_{\tau \sim p(\tau|\pi)} \sum_i R_{t+i}$ and the state-action value function $Q_t^\pi \triangleq \mathbb{E}_{\tau \sim p(\tau|\pi)} \sum_i R_{t+i}^{ext}$.

Under the condition of assumption $||(1 - \alpha_t) \cdot \widetilde{R}_t^{\text{int}}|| \le C$, we have

$$\hat{Q}_t^{\pi} = \mathbb{E}_{\tau \sim p(\tau | \pi)} \sum_i R_{t+i}$$

$$= \mathbb{E}_{\tau \sim p(\tau | \pi)} \sum_i [R_{t+i}^{\text{ext}} + (1 - \alpha_t) \cdot \widetilde{R}_t^{\text{int}}]$$

$$\le \mathbb{E}_{\tau \sim p(\tau | \pi)} \sum_i (R_{t+i}^{\text{ext}} + C)$$

$$= \mathbb{E}_{\tau \sim p(\tau | \pi)} \sum_i R_{t+i}^{\text{ext}} + \mathbb{E}_{\tau \sim p(\tau | \pi)} \sum_i C$$

$$= Q^{\pi} + (T - t)C$$

Similarly, we have

$$\hat{Q}_t^{\pi} \ge Q_t^{\pi} - (T - t)C.$$

Therefore, we have $Q_t^{\pi} - (T - t)C \le \hat{Q}_t^{\pi} \le Q_t^{\pi} + (T - t)C$.

Assuming under the state $\mathbf{s}_t$, the optimal action $\mathbf{a}_t^{\text{opt}}$, suboptimal action $\mathbf{a}_t^{\text{sub}}$ satify $Q(\mathbf{s}_t, \mathbf{a}_t^{\text{sub}}) < Q(\mathbf{s}_t, \mathbf{a}_t^{opt})$, we have

$$Q(\mathbf{s}_t, \mathbf{a}_t^{\text{opt}}) - (T - t)C \le \hat{Q}(\mathbf{s}_t, \mathbf{a}_t^{\text{opt}}) \le Q(\mathbf{s}_t, \mathbf{a}_t^{\text{opt}}) + (T - t)C$$
$$Q(\mathbf{s}_t, \mathbf{a}_t^{\text{sub}}) - (T - t)C \le \hat{Q}(\mathbf{s}_t, \mathbf{a}_t^{\text{sub}}) \le Q(\mathbf{s}_t, \mathbf{a}_t^{\text{sub}}) + (T - t)C$$

For the above formulaic simplification, we have

$$\hat{Q}(\mathbf{s}_t, \mathbf{a}_t^{\text{opt}}) - \hat{Q}(\mathbf{s}_t, \mathbf{a}_t^{\text{sub}}) \ge Q(\mathbf{s}_t, \mathbf{a}_t^{\text{opt}}) - Q(\mathbf{s}_t, \mathbf{a}_t^{\text{sub}}) - 2(T - t)C$$

Due to $C = \frac{Q(\mathbf{s}_t, \mathbf{a}_t^{\text{opt}}) - Q(\mathbf{s}_t, \mathbf{a}_t^{\text{sub}})}{2(T-t)}$, we know that $\hat{Q}(\mathbf{s}_t, \mathbf{a}_t^{\text{opt}}) - \hat{Q}(\mathbf{s}_t, \mathbf{a}_t^{\text{sub}}) > 0$. $\qquad \square$

