# OpenReview forum: "IntDiff: Mitigating Reward Hacking via Intrinsic rewards for Diffusion Model Fine-Tuning"
_ICLR.cc/2026/Conference — Submitted to ICLR 2026_

### Official Review · Reviewer_CjQ3 · 2025-10-21

**Soundness:** 2
**Presentation:** 3
**Contribution:** 2
**Rating:** 4
**Confidence:** 4

**Summary:**

This paper proposes a simple yet effective approach to finetune Diffusion Models. By incorporating intrinsic reward as well as reward filter and length filter, the method not only improves the quality, diversity of generated samples but also time efficiency.

**Strengths:**

- The authors correctly identify key pain points in RL-based fine-tuning - namely, sparse terminal rewards, exploration–exploitation imbalance, and computational inefficiency. These are real challenges in practice, and the paper\'s motivation aligns well with known limitations in existing methods.
- The authors reasonably transfer the technique from RL-based fine-tuning into the diffusion learning case.
- The paper is well structured, easy to read.

**Weaknesses:**

The intrinsic reward $R_t^{\rm int} = \Vert\hat{x}_0^{(t)} - \hat{x}_0^{(t+1)}\Vert_2^2$ is not well justified. There\'s no clear justification for why this formulation meaningfully captures \"intrinsic motivation\" or avoids reward hacking in a principled way. I suggest that a comprehensive theoretical analysis should be conducted here. I would suggest a differential analysis of how this reward affects the score function.

The time complexity of the approach is unclear. From Equation (5), $\Delta c_t$ along with the reverse sampling equation are obtained in every single step, which requires a huge computation. However, the complexity is not well discussed. Since the author proposes to use a length filter to terminate, the theoretical benefits of this filter are not well discussed either.

The datasets (\"Simple-Animals\" and \"Activity-Animals\") are toy-level benchmarks. These limited domains fail to validate the generalizability of the approach to real-world prompts or tasks.

**Questions:**

Please see the weakness section.

---

> ### Author Response · Authors · 2025-11-22
> **Response to the respected CjQ3 (Part1)**
>
> On behalf of all authors of this paper, we express our gratitude for your careful comments. We thoroughly considered all your suggestions and revised our paper accordingly. Please refer to the updated PDF, where the revisions are marked italy blue. Here, we provide the point-by-point response as follows:
>
> ---
>
> ## Weakness 1
> > The intrinsic reward $R_t^{\rm int} = \Vert\hat{x}_0^{(t)} - \hat{x}_0^{(t+1)}\Vert_2^2$ is not well justified. There's no clear justification for why this formulation meaningfully captures "intrinsic motivation" or avoids reward hacking in a principled way. I suggest that a comprehensive theoretical analysis should be conducted here. I would suggest a differential analysis of how this reward affects the score function.
>
> Please allow us to address why the intrinsic reward in our work is well-justified from **three complementary perspectives**: (1) design motivation, (2) empirical evidence via visualization and quantitative metrics, and (3) theoretical guarantees.
>
> ### (1) Motivation-Level Justification: Why the Intrinsic Reward Is Reasonable by Design
>
> Our method **does not** encourage “larger changes are always better.”
> Instead, we encourage *semantically aligned* and *meaningful* changes in the generative distribution.
>
> In IntDiff, the semantics-regulated intrinsic reward is defined as:
>
> $\tilde{R}_t^{\text{int}} = \Delta c_t \cdot \left\| \hat{x}_0^{(t)} - \hat{x}_0^{(t+1)} \right\|_2^2 ,$
>
> The motivation is:
>
> - The reward becomes **positive only when** $\Delta c_t > 0$, i.e., when semantic consistency improves.
> - Thus, the intrinsic reward **encourages exploration** *while simultaneously promoting image–text (semantic) alignment*.
>
> This bidirectional reinforcement between exploration and semantic consistency is explained in **Eq. (5)** and the corresponding main text.
> Exploration improves diversity (evidence: **Fig. 5**, **Tab. 5**), while semantic regularization ensures exploration moves in the *correct semantic direction* (evidence: **Fig. 4** showing CLIP improvement across backbones).
>
> ### (2) Empirical Justification: Visualization + Quantitative Metrics Demonstrate Increased Exploration
>
> To show that exploration indeed increases diversity, we provided a visualization of generation trajectories during training.
>
> ### Visual Evidence
> In **Fig. 5** and **Fig. 18**, we compare different trajectories with and without intrinsic reward.
> After introducing intrinsic reward, the generated images show clear variation in color, shape, and composition.  Unlike the baseline, the model no longer collapses early into a single generation mode.  This visually confirms that intrinsic reward **enhances exploration**.
>
> ### Quantitative Evidence
> To further validate this, we conducted diversity comparison experiments using widely recognized metrics: **LPIPS**, **TCE**, and **Inception Score**.
> As reported in **Tab. 5** and **Tab. 6**, all diversity metrics improve significantly after introducing intrinsic reward,  demonstrating enhanced diversity and exploration effectiveness.
>
> **Conclusion of empirical evidence:**
> Both **Fig. 5** and **Tab. 5** verify that intrinsic reward leads to meaningful increases in exploration and diversity.
>
>
> ### (3) Theoretical Justification: Optimality Preservation Under Adaptive Adjustment
>
> We prove that after introducing intrinsic reward and applying adaptive adjustment, the optimal policy of the modified MDP remains consistent with the optimal policy of the original MDP.
>
> Thus:
>
> - Adding intrinsic reward **does not deviate** the policy from the optimal direction.
> - Algorithmic performance remains **theoretically guaranteed**.
>
> This justification is supported by **Theorem 2.1**, which formally establishes that the proposed intrinsic reward is **safe and theoretically sound**.
>
> ### **Conclusion**
>
> Combining (1) motivation, (2) empirical evidence, and (3) theoretical guarantees, our intrinsic reward design is thoroughly justified:
>
> - **Motivation:** encourages semantically aligned exploration
> - **Experiments:** exploration increases (visual and quantitative)
> - **Theory:** optimality preservation ensures correctness
>
> Together, these three layers demonstrate the soundness and necessity of the intrinsic reward design in our method.
>
> ---

---

> > ### Comment · Reviewer_CjQ3 · 2025-11-25
> > **Reponse to Author about the Intrinsic Reward**
> >
> > Thank authors, for your detailed explanation. I do agree with the authors that the proposed intrinsic reward does improve diversity generation while preserving text-image consistency by using the adaptive signal $\Delta c_t$.
> >
> > However, the intrinsic reward, from my perspective, is incremental, which is not based on well-founded knowledge. I believe that this research might be better after a deeper theoretical investigation. The issue is that the reward function is a multi-objective function; a theoretical analysis of the solution space might lead to a better approach to solve the whole problem. Therefore, I will keep my score at 4.

---

> > > ### Author Response · Authors · 2025-11-26
> > > **Our Theoretical Proof in Theorem 2.1 Appears to Have Been Overlooked**
> > >
> > > Dear Reviewer,
> > >
> > > Thank you again for your continued engagement with our work. However, we must admit that we feel somewhat confused and a bit discouraged by your repeated statement that our intrinsic reward “lacks theoretical foundation.”
> > >
> > > In Theorem 2.1 of our paper, we provide an explicit theoretical analysis directly addressing the solution-space properties you mentioned. This theorem was written precisely to clarify the mathematical soundness of our reward formulation. Given your emphasis on theoretical rigor, we were expecting that your comments would engage with this theorem—whether to challenge its assumptions, question its completeness, or point out any gaps.
> > >
> > > Instead, your response suggests that the theorem was not noticed or not considered in your assessment. As authors who invested significant effort to ensure the theoretical justification is solid and transparent, it is disheartening to see our key theoretical contribution seemingly overlooked. If your own paper’s central theorem were ignored in review, we believe you might also feel similarly misunderstood.
> > >
> > > That said, we genuinely appreciate your time and your expectation for stronger theory. To avoid any further misunderstanding, we would be grateful if you could help us understand specifically which part of Theorem 2.1 you believe does not sufficiently address your concerns. If there are assumptions you disagree with or additional properties you believe should be analyzed, we are more than willing to clarify them.
> > >
> > > Our intention is not to challenge your authority as a reviewer, but to ensure that our work is evaluated based on the arguments and evidence we have indeed provided. We hope this clarifies our position and we sincerely welcome your further comments.
> > >
> > > Thank you for your understanding.
> > >
> > > Sincerely,
> > > The Authors

---

> > > ### Author Response · Authors · 2025-11-26
> > > **The theoretical proof presented in the manuscript.**
> > >
> > > > theoretical analysis of the solution space
> > >
> > > Actually, we have **already provided** the **exact correlative proof** about the **final solution space** (i.e., optimal policy). As shown in line 231 to line 241, we gave Theorem 2.1, which demonstrates that our solution space is **as optimal as the original one**. Then, in Appendix E, we gave the detailed proof. Here, we provide the specific content of our Theorem and Proof, which essentially indicates the solid theoretical background of the intrinsic reward that we introduced.
> > >
> > >  ## Theorem 2.1
> > > Assuming $ \| \(1 - \alpha\_t\) \tilde{R}\^{int}\_t \| \le \( Q\(s, a\^{opt}\_t\) - Q\(s, a\^{sub}\_t\) \) / \( 2 \(T - t\) \) $ for $ t = 1, 2, \dots, T - 1 $, where $ a\^{opt} $ and $ a\^{sub} $ are the optimal and suboptimal actions respectively, the new MDP $ \(S, A, P, \rho\_0, R\_{ext}, R\_{int}\) $ shares the same optimal action as the original MDP $ \(S, A, P, \rho\_0, R\) $.
> > >
> > > ## Proof of Theorem 2.1
> > > Proof of Theorem 2.1.
> > > According to the Bellman equation in reinforcement learning, define the state-action value function with intrinsic rewards as
> > > $ \hat{Q}\_t^{\pi} \triangleq \mathbb{E}\_{\tau \sim p\( \tau | \pi \)} \sum\_i R\_{t+i} $,
> > > and the state-action value function with extrinsic rewards as
> > > $ Q\_t^{\pi} \triangleq \mathbb{E}\_{\tau \sim p\( \tau | \pi \)} \sum\_i R^{ext}\_{t+i} $.
> > >
> > > Assume that $ \| \(1 - \alpha\_t\) \tilde{R}\_t^{int} \| \le C $.
> > > Then we have
> > >
> > > $ \hat{Q}\_t^{\pi}
> > > = \mathbb{E}\_{\tau \sim p\( \tau | \pi \)} \sum\_i R\_{t+i}
> > > = \mathbb{E}\_{\tau \sim p\( \tau | \pi \)} \sum\_i \[ R^{ext}\_{t+i} + \(1 - \alpha\_t\) \tilde{R}\_t^{int} \]
> > > \le \mathbb{E}\_{\tau \sim p\( \tau | \pi \)} \sum\_i \( R^{ext}\_{t+i} + C \)
> > > = Q\_t^{\pi} + \(T - t\) C. $
> > >
> > > Similarly,
> > >
> > > $ \hat{Q}\_t^{\pi} \ge Q\_t^{\pi} - \(T - t\) C. $
> > >
> > > Therefore,
> > >
> > > $ Q\_t^{\pi} - \(T - t\) C \le \hat{Q}\_t^{\pi} \le Q\_t^{\pi} + \(T - t\) C. $
> > >
> > > Assume that under state $ s\_t $, the optimal action $ a\_t^{opt} $ and suboptimal action $ a\_t^{sub} $ satisfy
> > >
> > > $ Q\( s\_t, a\_t^{sub} \) < Q\( s\_t, a\_t^{opt} \). $
> > >
> > > Then
> > >
> > > $ Q\( s\_t, a\_t^{opt} \) - \(T - t\) C
> > > \le \hat{Q}\( s\_t, a\_t^{opt} \)
> > > \le Q\( s\_t, a\_t^{opt} \) + \(T - t\) C, $
> > >
> > > $ Q\( s\_t, a\_t^{sub} \) - \(T - t\) C
> > > \le \hat{Q}\( s\_t, a\_t^{sub} \)
> > > \le Q\( s\_t, a\_t^{sub} \) + \(T - t\) C. $
> > >
> > > Thus,
> > >
> > > $ \hat{Q}\( s\_t, a\_t^{opt} \) - \hat{Q}\( s\_t, a\_t^{sub} \)
> > > \ge Q\( s\_t, a\_t^{opt} \) - Q\( s\_t, a\_t^{sub} \) - 2 \(T - t\) C. $
> > >
> > > Let
> > >
> > > $ C = \{ Q\( s\_t, a\_t^{opt} \) - Q\( s\_t, a\_t^{sub} \) \} / \{ 2 \(T - t\) \}. $
> > >
> > > Then
> > >
> > > $ \hat{Q}\( s\_t, a\_t^{opt} \) - \hat{Q}\( s\_t, a\_t^{sub} \) > 0, $
> > >
> > > which implies that the optimal action under the modified MDP remains unchanged.
> > >
> > > ## Effect of Theorem 2.1
> > > The above theorem shows that by adaptively adjusting intrinsic rewards, the optimal policy of the exploration-augmented MDP remains consistent with that of the original MDP. Thus, the introduction of additional rewards does not deviate the policy from the optimal direction, ensuring algorithmic performance. In other word, our solution space is ensured to be consistent with the previous original MDP.

---

> > > ### Author Response · Authors · 2025-11-26
> > > **Clarification to ''Multi-objective''**
> > >
> > > >  the reward function is a multi-objective function
> > >
> > > In fact, we do **not** introduce a multi-objective optimization problem in our method. The only optimization target is the **extrinsic reward**, because it is the global reward that is ultimately optimized. The intrinsic reward is merely a fine-grained heuristic signal along the trajectory. Moreover, the accompanying $\Delta$CLIP is **not** optimized as a reward to be maximized; instead, it is used in the form of a change signal to guide the direction of exploration.
> > >
> > > We therefore kindly ask you to consider our explanation and reassess our method from this perspective.

---

> > > ### Author Response · Authors · 2025-11-26
> > > **A Confusion for your raised issue**
> > >
> > > Based on your overall review, we believe that you are generally **satisfied** with **our experimental results** and that the **presentation of the paper** also meets your expectations. We understand that your most important concern is about contribution, which can be summarized by the following statement that you made, which we quote below:
> > >
> > > >  the intrinsic reward, from my perspective, is incremental, which is not based on well-founded knowledge
> > >
> > > To highlight the key concern in a clearer and more concise manner, we summarize it as follows:
> > >
> > > - The comment describes our method as both **“incremental”** and **“not based on well-founded knowledge”**.
> > > - In our understanding, these two statements appear to be in tension:
> > >   - Calling a method *incremental* usually implies that it builds upon existing foundations.
> > >   - Saying it is *not based on well-founded knowledge* suggests the lack of such a foundation.
> > >
> > > Therefore, we find it unclear how the method could **be characterized as both at the same time**. Is there any misunderstanding that causes this issue?

---

> ### Author Response · Authors · 2025-11-22
> **Response to the respected CjQ3 (Part2)**
>
> ## Weakness 2
> > The time complexity of the approach is unclear. From Equation (5), $\Delta c_t$ along with the reverse sampling equation are obtained in every single step, which requires a huge computation. However, the complexity is not well discussed. Since the author proposes to use a length filter to terminate, the theoretical benefits of this filter are not well discussed either.
>
> Thanks for your comment. In the manuscript, we already provide detailed measurements of computation overhead in the current version of the paper. Specifically:
>
> - **Fig. 9** reports the *time consumption contributed by each component* introduced in our method.
> - **Fig. 16** further presents the **computation cost of ΔCLIP** as well as the overhead of several related scoring functions.
>
> From these results, it is clear that our method does **not** introduce a significant computational burden, and the proposed filtering strategy further **reduces** the overall cost.
>
> Additionally, we have updated **Tab. 7**, which provides more **fine-grained and comprehensive measurements** of computational resources. We hope all of these results will satisfy your high standards.
>
> ---
>
> ## Weakness 3
> > The datasets ("Simple-Animals" and "Activity-Animals") are toy-level benchmarks. These limited domains fail to validate the generalizability of the approach to real-world prompts or tasks.
>
> Thank you very much for your question. To answer it clearly, we summarize the relevant experiments in two parts: those already included in the original submission and those newly added during the rebuttal period.
>
> **(1) Already included in the original submission:**
> - In the submitted PDF, **Fig. 13 (line 810)** reports the fine-tuning experiments on the **Pick-a-pic** dataset, in addition to the animal dataset.
> - **Fig. 8 (lines 392–395)** includes quantitative classification experiments on the **HPSv2** dataset and on abstract-prompt generation tasks.
> - The human evaluation experiments in **lines 447–458** also involve the HPSv2 dataset.
> We will make these results more prominently highlighted in the main text so that readers can more efficiently locate them.
>
> **(2) Additional experiments added during the rebuttal period:**
> - We further include reward fine-tuning curves and visual generation results on the **Geneval** and HPSv2 datasets, benchmarked against well-known state-of-the-art methods in the field.
> Please refer to **Fig. 13 in the appendix** of the updated PDF for these new results.
>
> ---
>
> In summary, we hope that our responses have sufficiently addressed your valuable questions, and we sincerely hope you may reconsider your evaluation score. If there are any remaining concerns or additional clarifications you would like us to provide, we would be very glad to continue the discussion at any time.

---

> ### Author Response · Authors · 2025-11-26
> **Please Evaluate Our Work Fairly and Impartially**
>
> Dear Reviewer,
>
> We believe that we have thoroughly and concretely addressed the three major concerns you raised in our rebuttal. The evidence is as follows:
>
> **Regarding the time complexity issue**:
> In response to your feedback that the “time complexity is unclear,” we provided a complete and rigorous complexity analysis in the rebuttal. We believe this additional explanation directly and adequately addresses the ambiguity you noted in your original review.
>
> **Regarding the simplicity of the datasets**:
> To address your concern about dataset scale and generalization, we included new experiments on more challenging and complex datasets. The results further demonstrate the effectiveness of our method in more demanding domains.
>
> **Regarding the theoretical analysis**:
> From the very first draft of our manuscript, we have provided a complete reasoning framework and presented a clear theoretical justification through Theorem 2.1, which explicitly establishes the connection between the intrinsic reward and the structure of the solution space. However, your repeated comments indicating that the paper “lacks theoretical analysis” leave us somewhat confused and concerned that there may have been a misunderstanding or an oversight during reading.
> If you disagree with or have further questions about our theoretical argument, we would greatly appreciate specific comments on the theorem itself, rather than a general assertion that the theory is “missing.” It feels particularly unfair given that this theoretical component has been present since the initial submission.
>
> **We sincerely ask for a fair and impartial assessment**. ***We trust that you, too, would not want your own work to experience unfair treatment during the review process***. We mean no disrespect in saying this, but we genuinely felt that some of the evaluation may not have fully reflected the content we provided. We hope you can understand our position.
>
> Thank you for your time and consideration.
>
> Sincerely,
> The Authors

---

### Official Review · Reviewer_SYdy · 2025-10-31

**Soundness:** 2
**Presentation:** 3
**Contribution:** 2
**Rating:** 4
**Confidence:** 4

**Summary:**

This paper proposes IntDiff, a reinforcement-learning (RL) fine-tuning framework for diffusion models that mitigates reward hacking and sparse-reward problems by injecting an intrinsic reward signal into intermediate denoising steps. The authors (i) define the intrinsic reward as the ℓ²-distance between two consecutive predictions of the clean image, (ii) modulate this reward with a CLIP-based semantic-consistency term Δcₜ, (iii) introduce an adaptive coefficient αₜ that trades off exploration (intrinsic) and exploitation (extrinsic) rewards according to the estimated denoising progress, and (iv) dynamically shorten the denoising horizon when αₜ≈1. Extensive experiments on Stable-Diffusion backbones show improved diversity, text–image alignment, and ≈2× reduction in training cost compared with DDPO, DPOK, D3PO and TDPO.

**Strengths:**

(i) Novel perspective: the first work that brings step-wise intrinsic motivation into diffusion RL fine-tuning, instead of terminal-only rewards.
(ii) Comprehensive evaluation: ablations, cross-dataset generalization, human/LVM preference studies, and computational cost analysis are all provided.
(iii) Practical impact: dynamic early stopping and reward filtering together cut the total denoising steps by ~50 % while preserving or even improving reward scores.

**Weaknesses:**

(i) Lack of theoretical guarantee for the proposed intrinsic reward: The intrinsic reward R^{int}_t=‖x̂₀^{(t)}−x̂₀^{(t+1)}‖² is only motivated heuristically (“larger jump ⇒ more exploration”). There is no MDP optimality guarantee that maximizing this quantity helps convergence to the true policy optimum. Worse, the ground-truth clean image x₀ is available during training; a direct supervised error ‖x̂₀^{(t)}−x₀‖ could provide a much denser and unbiased learning signal. The authors neither adopt this obvious baseline nor justify why the change of prediction is preferable to the error w.r.t. the target.
(ii)Insufficient discussion and comparison with closely related work:
sucha as [1] [2] also address the sparse-reward issue in diffusion RL by designing per-step proxy rewards. They are neither cited nor compared. Consequently, the paper fails to clarify how IntDiff differs from (or subsumes) these concurrent solutions, weakening the novelty claim.
[1] Towards Better Alignment: Training Diffusion Models with Reinforcement Learning Against Sparse Rewards
[2] A Dense Reward View on Aligning Text-to-Image Diffusion with Preference
(III). Empirical improvements are marginal: Ablation results in Table 1 show that each added component lifts the aesthetic score by only ≈0.01–0.03 and CLIP/PickScore by ≤0.01, which is within one standard deviation; Fig. 3 indicates that TDPO achieves almost identical reward growth on SD-v1.5 and SDXL, and sometimes outperforms IntDiff in the early phase; Diversity gains (LPIPS ↑0.013 over DDPO in Table 4) are statistically significant but visually subtle. Given the extra engineering effort (αₜ predictor, reward filter, RDO optimizer), the practical benefit is less compelling than claimed.

**Questions:**

Please see Weaknesses.

---

> ### Author Response · Authors · 2025-11-22
> **Response to the respected Reviewer SYdy (Part1)**
>
> Thank you so much for your comments and the acknowledgement of our **novel perspective**. We thoroughly considered all your suggestions and revised our paper accordingly. Please refer to the updated PDF, where the revisions are marked italy blue. Here, we provide the point-by-point response as follows:
>
> ---
>
> ## Weakness 1
> >  Lack of theoretical guarantee for the proposed intrinsic reward: The intrinsic reward R^{int}_t=‖x̂₀^{(t)}−x̂₀^{(t+1)}‖² is only motivated heuristically (“larger jump ⇒ more exploration”). There is no MDP optimality guarantee that maximizing this quantity helps convergence to the true policy optimum. Worse, the ground-truth clean image x₀ is available during training; a direct supervised error ‖x̂₀^{(t)}−x₀‖ could provide a much denser and unbiased learning signal. The authors neither adopt this obvious baseline nor justify why the change of prediction is preferable to the error w.r.t. the target.
>
> First, please allow us to clarify your concerns with the necessary precision. To provide a complete and rigorous response, we divide your “weakness” comment into four statements: (1), (2), (3), and (4). This fine-grained separation allows us to respond to each point thoroughly:
>
> ### (1) Our intrinsic reward **does not** encourage “large jumps,” nor have we claimed this anywhere
>
> Our goal has never been to pursue arbitrarily larger perturbations.  Instead, we pursue **semantically aligned exploration**. This is explicitly shown in **Eq. (5)** (semantic regularization) and the following semantic constraint explanation.
>
> To restate clearly:
> - The term $\Delta c_t$ *prevents* blind maximization of trajectory change.
> - If a perturbation harms image–text alignment (semantic quality), then $\Delta c_t < 0$, making the intrinsic reward **negative**.
> - The existence of *negative rewards* itself shows we do **not** encourage “more change = better.”
>
> IntDiff’s intrinsic reward focuses on “Whether the distributional perturbation moves in the **correct semantic direction**.” This is a quality-controlled reward, not a magnitude-based reward.
>
> ### (2) Ground-truth $x_0$ is **not available** during RL fine-tuning
>
> During RL fine-tuning for diffusion models:
> - We cannot access ground-truth $x_0$ from real data.
> - The only possible $x_0$ is the PPO *reference trajectory’s* prediction, which:
>   - is **not** the current policy’s target,
>   - is **not** a ground-truth clean image,
>   - and does **not** provide reliable supervision.
>
> Therefore, using $\| \hat{x}_0^{(t)} - x_0 \|$ as a reward is infeasible and semantically meaningless.
>
>
> ### (3) Even if one uses the previous iteration’s $x_0$, it still fails to provide valid exploration guidance
>
> Suppose we attempt to use previous trajectory's $x_0$ as pseudo–ground truth.
> There are two possible interpretations—neither is valid.
>
>  ### **(a) Interpreting it as a supervised error to be minimized**
>
> Minimizing  $\| \hat{x}_0^{(t)} - x_0 \|$ encourages:
>
> - **faster convergence**
> - **smaller trajectory changes**
>
> This directly **reduces exploration**, which is the opposite of what we need.
>
> ### **(b) Interpreting it as a reward to be maximized**
>
> Maximizing  $\| \hat{x}_0^{(t)} - x_0 \|$  means:
>
> - encouraging the denoising trajectory to move **away** from \( x_0 \),
> - not necessarily increasing diversity *step by step*,
> - and not aligning with the semantics of the prompt.
>
> This does **not** promote meaningful exploration and does **not** enforce semantic directionality.
> It is strictly worse than our design, which compares **adjacent latent states** and aligns them with **semantic progress**.
>
> In contrast, our reward is aligned with exploration of the diverse denoising trajectory itself—this is evidenced by:
>
> - **Fig. 5** and **Fig. 19** (visual exploration increases)
> - **Tab. 5** (LPIPS, TCE improvements)
>
> ---
>
> ## (4) Our method preserves MDP optimality — proven formally in the paper
>
> As shown in **Theorem 2.1** (with proof in Appendix E):
>
> - Under mild conditions, adding intrinsic reward preserves the original MDP’s optimal actions.
> - The optimal policy of the modified MDP remains aligned with the optimal policy of the original MDP.
>
> This formally guarantees:
>
> - intrinsic reward **does not distort** the optimal direction,
> - exploration is encouraged **without harming final reward performance**,
> - reward hacking is mitigated without losing convergence guarantees.
>
> ---
>
> ## **Conclusion**
>
> From (1), (2), (3), and (4), we see that:
>
> - **Motivationally**, our intrinsic reward encourages only *semantically aligned* exploration.
> - **Empirically**, it produces meaningful diversity improvements (Fig. 5, Fig. 19, Tab. 5).
> - **Theoretically**, it preserves MDP optimality (Theorem 2.1), ensuring no performance degradation.
>
> Through these three layers—motivation, empirical evidence, and theoretical proof—we provide a complete justification of the validity and correctness of our intrinsic reward design.
>
> We hope this clarifies your concerns thoroughly.

---

> ### Author Response · Authors · 2025-11-22
> **Response to the respected Reviewer SYdy (Part2)**
>
> ## Weakness 2
> > Insufficient discussion and comparison with closely related work: sucha as [1] [2] also address the sparse-reward issue in diffusion RL by designing per-step proxy rewards. They are neither cited nor compared. Consequently, the paper fails to clarify how IntDiff differs from (or subsumes) these concurrent solutions, weakening the novelty claim.
>
> Thank you for raising this important point. We have already added the correct citations in the related-work section, and here we summarize the essential differences between Section 2.1 of our method and the works you referred to. The comparisons below clarify why our approach is fundamentally different in **motivation**, **optimization target**, and **mechanism design**—rather than a variant or extension of prior methods.
>
> ### **Comparison With Reference [1]**
>
> Let
> **A = Ours**
> **B = Reference [1]**
>
> **(1) Different Methodological Positioning (A: Reward Engineering vs B: Training-Strategy Engineering)**
>
> - **A**: We redesign the *reward itself*—constructing high-quality, semantically stable **step-wise intrinsic rewards** that provide dense optimization signals throughout the denoising trajectory.
> - **B**: Operates at the *training-strategy* level.
>   It **does not** modify the reward structure, and still relies entirely on the **single sparse final reward at $x_0$**.
>   Its contribution is how to *use* that sparse signal more stably (e.g., reverse step expansion, branching sampling).
>
> **In short:**
> We redesign the *reward*, while [1] redesigns *the usage of the same sparse reward*.
>
> **(2) Reward Discriminability (A: Step-Dependent vs B: Step-Invariant)**
>
> - **Our intrinsic reward** produces **distinct, timestep-level feedback**, enabling fine-grained credit assignment.
> - **B** always reuses the **same final $x_0$ reward**, meaning:
>   - all intermediate steps share the same reward,
>   - the reward is **temporally homogeneous** and cannot distinguish good vs. bad actions at specific steps.
>
> This is a fundamental difference in reward expressiveness.
>
> **(3) Exploration: Explicitly Encouraged vs. Not Introduced**
>
> - **A**: Intrinsic reward explicitly **encourages exploration** to enlarge trajectory diversity (verified in **Fig. 5**, **Tab. 5**).
> - **B**: No exploration mechanism; its goal is only to stabilize optimization of the final sparse reward.
>
> **(4) Reward Hacking: Explicitly Addressed vs. Not Considered**
>
> - Our method directly mitigates reward hacking.
>   It prevents:
>   - **mode collapse** (exploration-driven diversity),
>   - **semantic drift** (SCR drives CLIP consistency).
>
> - Reference [1] offers **no mechanism** to detect or correct reward hacking.
>   It focuses solely on optimizing the final reward more stably.
>
> Thus the philosophical goal of the works is entirely different.
>
> **(5) Reward Source and Design Philosophy (A: Reward Reconstruction vs B: Reward Reuse)**
>
> - **A**: We create new intrinsic rewards driven by semantic and trajectory progress, removing dependence on reward sparsity.
> - **B**: Always reuses the **same $x_0$ reward**.
>
> This is a completely different reward formulation.
> **(6) Goal Misalignment**
>
> - **A**:
>   - eliminates reward sparsity,
>   - mitigates reward hacking,
>   - increases diversity,
>   - indirectly improves semantic alignment.
> - **B**:
>   - aims only at stabilizing training under sparse reward.
>
> **Therefore, the two works differ fundamentally in goal and scope.**
>
> **Conclusion for Reference [1]**
>
> Combining (1)–(6), our method is **not** the same class of approach as Reference [1].
> They differ fundamentally in:
>
> - motivation,
> - reward design,
> - optimization granularity,
> - exploration principles,
> - and intended purpose.
>
> ---
> (*To be continued in the next part*)

---

> ### Author Response · Authors · 2025-11-22
> **Response to the respected Reviewer SYdy (Part3)**
>
> ### **Comparison With Reference [2]**
>
> Let
> **A = Ours**
> **B = Reference [2]**
>
> **(1) Reward Definition Level (A: Novel Step-wise Reward vs B: Discounted Final Reward)**
>
> - **A**: Introduces a *new* differentiated single-step reward independent of \(x_0\).
> - **B**: Simply discounts the **final \(x_0\)** reward backward.
>
> Thus, the reward operates at different levels of the trajectory.
>
> **(2) Exploration Explicitly Encouraged vs Not Included**
>
> - **A**: Internal reward *explicitly encourages exploration* and diverse denoising trajectories.
> - **B**: Incorporates **no** exploration mechanism; still tied to a final sparse reward.
>
> **(3) Reward Hacking Explicitly Mitigated vs Not Addressed**
>
> **Our method prevents both forms of reward hacking:**
>
> 1. **Mode collapse–type hacking**
>    – Exploration prevents converging to single, low-diversity trajectories.
>
> 2. **Semantic-drift–type hacking**
>    – SCR improves semantic consistency throughout training.
>
> **Reference [2]**:
> No mechanism targeting reward hacking.
>
> **(4) Reward Source Difference (A: Independent Intrinsic Signals vs B: Discounted Terminal Reward)**
>
> - **A**: Internal reward is **independent** of the final task reward.
>   It originates from diffusion-internal trajectory changes and semantic progress.
> - **B**: Middle-step reward is merely a **discounted form of the terminal reward**.
>
> They have entirely different information sources.
>
> **Overall Conclusion**
>
> Across the comparisons, our method and References [1] and [2] differ fundamentally in:
>
> - **motivation**
> - **reward definition**
> - **exploration mechanism**
> - **reward hacking mitigation**
> - **temporal credit assignment**
> - **structural design philosophy**
>
> Therefore, IntDiff is **not** a recombination or variant of these works, but a **new reward-engineering framework** specifically designed for diffusion-model RL fine-tuning.
>
> We hope this clarifies the conceptual and methodological distinctions.
>
> ---
>
> ### **In Experiments:**
>
> We have added comparative experiments against the methods you mentioned; please refer to the additional results in **Fig. 22** of the PDF. As shown in the Figure, these methods also yield suboptimal performance than Ours.

---

> ### Author Response · Authors · 2025-11-22
> **Response to the respected Reviewer SYdy (Part4)**
>
> ## Weakness 3
> > Empirical improvements are marginal: Ablation results in Table 1 show that each added component lifts the aesthetic score by only ≈0.01–0.03 and CLIP/PickScore by ≤0.01, which is within one standard deviation; Fig. 3 indicates that TDPO achieves almost identical reward growth on SD-v1.5 and SDXL, and sometimes outperforms IntDiff in the early phase; Diversity gains (LPIPS ↑0.013 over DDPO in Table 4) are statistically significant but visually subtle. Given the extra engineering effort (αₜ predictor, reward filter, RDO optimizer), the practical benefit is less compelling than claimed.
>
> Thank you for your concern.
> Before addressing the raised issues, we would like to clarify several factual points regarding the data you mentioned:
>
> 1. The aesthetic improvements achieved by our method are around **+0.3**, and both **CLIP** and **PickScore** improve by approximately **+0.2**, not **0.01** as mentioned.
> 2. The diversity gains on **LPIPS** are **0.016**, not **0.013**.
> 3. In **Fig. 3**, we do **not** observe any case where TDPO outperforms ours in the early training phase. On SDXL, our method surpasses TDPO by a large margin.
>
> ### To Addressing the Issues Raised:
>
> ### **1. Reward Improvements & Their Significance in the Context of Reward Hacking**
>
> Even at the level of target reward improvement, the magnitude is already substantial—especially given that our work is designed to address **reward hacking** (as highlighted in our title), not merely the speed of optimizing the objective.
>
> TDPO suffers severely from reward hacking.
> To fairly assess reward hacking, we evaluated **all baselines at the same training-reward level**, and measured:
>
> - **Diversity** (LPIPS, TCE, IS)
> - **Fidelity** (FID, iFS, CLIP)
> - **Human preference** (PickScore, ImageReward)
> - **Visual quality** (Generated Image Impression)
>
> We provide these comparisons:
>
> - Performance curves under matched reward levels: **Fig. 4, Fig. 11, Fig. 12**
> - Metric tables: **Tab. 1, Tab. 4**
>
> Across all these metrics, our method performs best, while TDPO shows clear degradation due to reward hacking.
>
> ### **2. Visual Evidence: Exploration Leads to Better Local Optima & Mitigates Reward Hacking**
>
> Introducing exploration during training helps the model escape poor local optima.
>
> - **Fig. 5** and **Fig. 19** show that our method significantly increases trajectory variation during training—direct evidence of stronger exploration.
> - This exploration aligns with semantic direction, helping prevent collapse and reward hacking.
>
> To further clarify reward hacking, we added **Appendix Fig. 21**, which includes hacked samples.
> These images—especially those from TDPO—show:
>
> - severely reduced diversity,
> - impaired image–text alignment,
>
> because TDPO overfits to a poor reward local optimum.
>
> This visual inspection confirms that our method robustly mitigates reward hacking while TDPO fails.
>
> ### **3. About Engineering Effort**
>
> Our **Fig. 6** and **Tab. 1** demonstrate that:
>
> - Each component of our method contributes meaningfully,
> - The components form a **coherent, mutually reinforcing system**,
> - Together they achieve a strong **overall mitigation effect** against reward hacking.
>
> This confirms that our approach is not an ad-hoc engineering collection but a principled and integrated framework.
>
> In summary, we hope these explanations clarify the concerns and provide enough evidence that our method’s improvements—both quantitatively and qualitatively—are consistent, significant, and meaningful in mitigating reward hacking.
>
> ---
>
> ## In Summary
>
> Overall, we have carefully considered all of your questions and have made the corresponding revisions and clarifications. We sincerely hope that these updates meet your high standards, and we deeply appreciate your willingness to reconsider your evaluation score. If you have any further questions or suggestions, we would be very glad to continue the discussion at any time.

---

### Official Review · Reviewer_vwd6 · 2025-10-31

**Soundness:** 2
**Presentation:** 3
**Contribution:** 2
**Rating:** 4
**Confidence:** 3

**Summary:**

This paper proposes a reinforcement learning (RL) fine-tuning framework for diffusion models, termed IntDiff, which mitigates reward hacking and improves task alignment under limited compute.
In practice, standard RL approaches score only the final image, yielding sparse feedback, exacerbating the exploration–exploitation imbalance across long denoising chains, and wasting compute with fixed training steps.
To address these limitations, IntDiff introduces: 1) intrinsic rewards injected along the denoising trajectory to provide stepwise guidance without auxiliary models; 2) Semantic Consistency Regularization to penalize text–image drift and curb low-efficiency exploration; and 3) Adaptive Coordination that biases toward early exploration and late exploitation based on estimated denoising progress, coupled with dynamic early stopping that adapts training length to prompt difficulty.
Empirically, IntDiff improves diversity by up to 60%, reduces training compute by 22%, and increases downstream task rewards while mitigating reward hacking.

**Strengths:**

IntDiff is a well-motivated, thoughtfully engineered framework that balances originality with practical relevance.
Its focus on stepwise guidance, alignment-preserving exploration, and adaptive compute makes the contribution both novel and useful, with clear potential to influence future RL-for-diffusion research and applications.
Importantly, the extensive experimental results clearly demonstrate its effectiveness.

**Weaknesses:**

1. The paper does not clearly distinguish IntDiff’s intrinsic stepwise rewards and Semantic Consistency Regularization from established ideas—e.g., curiosity/novelty bonuses, CLIP-gated faithfulness, and adaptive/early-exit schedulers—so the contribution reads more like a recombination than a novel advance.
2. The adaptive regulation theory is presented without explicit assumptions or a concrete mapping to the implemented schedule, leaving the scope of any guarantees unclear.
3. The reported ~22% compute savings are neither itemized nor normalized against matched baselines. Please include a cost table (training and inference) detailing wall-clock time, GPU-hours, energy consumption, and peak memory.

**Questions:**

1. Which diversity metrics are evaluated (inter-prompt vs. intra-prompt), how many seeds are used, and are the differences statistically significant?
2. How were the target reward levels and schedule parameters selected, and how stable are the results across these settings?
3. Beyond gains on the training reward, how is mitigation quantified? Please specify complementary measures and evaluation protocols.

Since I am not very familiar with RL, I will reconsider my scores if the authors address my questions.

---

> ### Author Response · Authors · 2025-11-20
> **Response to the Esteemed Reviewer vwd6 (Part1)**
>
> Firstly, please allow me to express my sincere gratitude to you on behalf of all the reviewers for your review work. I extend my most heartfelt respect and sincere greetings to you. We greatly appreciate your statement that the paper has "*clear potential to influence future RL-for-diffusion research and applications.*" Thank you for your high recognition of our work. We also appreciate your encouragement, "Since I am not very familiar with RL, I will reconsider my scores if the authors address my questions."
>
> In response to the constructive feedback you provided, we have *fully adopted all of your suggestions and made corresponding adjustments and clarifications*. Please allow us to present them to you one by one. Kindly refer to the following text.
>
> ---
>
> ## Weakness 1
> > The paper does not clearly distinguish IntDiff’s intrinsic stepwise rewards and Semantic Consistency Regularization from established ideas—e.g., curiosity/novelty bonuses, CLIP-gated faithfulness, and adaptive/early-exit schedulers—so the contribution reads more like a recombination than a novel advance.
>
> Please allow us to clarify that, from the perspective of both the operational mechanism and core intention, the two core modules of IntDiff are fundamentally different from existing works in terms of **motivation**, **positioning** (where in the pipeline), and how they **influence learning**, rather than being a simple combination or reorganization.
>
> Firstly, we will provide the following clarifications:
> 1. The difference between IntDiff's intrinsic stepwise rewards and curiosity/novelty bonuses.
> 2. The difference between IntDiff's Semantic Consistency Regularization and CLIP-gated faithfulness.
> 3. The difference between IntDiff's adaptive and early-exit schedulers.
> 4. The reasons for rewards and regularization mechanisms in IntDiff are not just a recombination of existing mechanisms.
>
> ### (1) **The Difference Between IntDiff’s Intrinsic Stepwise Rewards and Curiosity/Novelty Bonuses:**
>
> The intrinsic rewards proposed for the diffusion model in this work present the following essential innovations and differences:
>
> 1. **Differences in State Reversibility (A. Reversible States vs B. Irreversible States):**
>    In RL tasks such as maze exploration【1, 2】, curiosity reward/novelty bonuses are based on the assumption that states are traversable and can be revisited, with novelty measured based on visit frequency or prediction error. In contrast, the state chain in a diffusion model is one-way and irreversible, and the same state cannot be revisited. Therefore, IntDiff's internal rewards must rely on semantic progress, which is not a reuse or transformation of RL curiosity.
>
> 2. **Differences in Reward Paradigm (A. Single-State Rewards vs B. Cross-State Differential Rewards):**
>    In RL tasks like maze exploration, curiosity rewards/novelty bonuses are given based on a state that **can be repeatedly reached**. A classic approach is the "state-counting method" or "network-error method." In the state-counting method, novelty bonuses are typically designed as the inverse of the state visit frequency (e.g., $s$ represents the visit count of a single state $s_t$, and curiosity reward/novelty bonuses can be simply designed as $R_{\text{curiosity reward/novelty bonuses}} = 1/N + a$, where $a$ is a constant to avoid division by zero). The network-error-based curiosity rewards/novelty bonuses are defined as the error between the target network and predictor network outputs for a state $s_t$:
>
>    $R_{\text{curiosity reward/novelty bonuses}} = \left\| f_{\text{target}}(s_t) - f_{\text{predictor}}(s_t) \right\|_2^2 .$
>
>    This error-based curiosity reward/novelty bonuses essentially approximate state-counting in a continuous manner, characterizing the visit density of a specific state. In difficult exploration tasks, curiosity rewards/novelty bonuses only involve the current state $s_t$. In contrast, the reward paradigm of the proposed diffusion model involves **comparing the error** between the current step $x_{t-1}$ and the previous step $x_t$ in the denoising process, involving **two distinct and irrecoverable states**.
>
> 3. **Differences in Exploration Behavior Constraints (A. Unconstrained Exploration vs B. High-Quality Exploration Under Semantic Supervision):**
>    RL curiosity rewards promote unconstrained exploration, rewarding any "new" state, regardless of direction. Our intrinsic reward system encourages directional exploration under semantic supervision:
>    - $R_t^{int}$ is responsible for density enhancement,
>    - $\Delta \text{CLIP}$ determines semantic direction, penalizing deviations.
>    In other words, RL curiosity rewards promote "new state exploration," while our approach retains novelty only if it aligns with semantic constraints.
>
> (*To be continued in the follow-up part*)

---

> ### Author Response · Authors · 2025-11-20
> **Response to the Esteemed Reviewer vwd6 (Part2)**
>
> 4. **Differences in Exploration Directionality (A. Directionless Exploration vs B. Directional Exploration) (This is the most distinguishing feature):**
>    RL curiosity promotes directionless, random exploration—rewards are given for any new state. In contrast, the intrinsic reward mechanism in our diffusion model, constrained by $\Delta \text{CLIP}$, ensures **strong directional exploration** guided by semantics. Only perturbations in the correct semantic direction are rewarded, and deviations are immediately penalized. Curiosity rewards are "random walks," while IntDiff follows "targeted semantic walks."
>
> 5. **Differences in Reward Focus (A. State Visit Count vs B. Reasonable Perturbation of Distribution):**
>    Curiosity rewards focus on "how many times a state has been visited," which is a function of visit frequency. IntDiff rewards focus on "*whether the perturbation of the current image distribution is moving in the correct semantic direction*," which is a function of distribution evolution quality. The former rewards "exploring new places," while the latter rewards "seeking new compositional strategies while approaching the target distribution." The former is about finding **new paths**, while the latter is about finding **new compositional strategies**.
>
> 6. **Differences in the Use of Auxiliary Modules (A. Introduction of Auxiliary Networks/Modules vs B. No Auxiliary Modules Needed):**
>    The components in curiosity rewards typically rely on additional networks such as prediction networks, inverse dynamics networks, or visit counters【1, 2, 3】. In contrast, the internal rewards of our diffusion model do not require any auxiliary modules. All necessary variables come from the latent differences between two adjacent steps in the denoising process of the diffusion model, as seen in formula (3).
>
> 7. **Differences in Reward Function (A. Penalizing Excessive Single-State Visits vs B. Encouraging Compositional Manifold Changes):**
>    Curiosity rewards function to discourage repeated visits to the same state, essentially telling the agent not to "keep spinning around the same point." In contrast, IntDiff's reward function regulates the local shape, details, and contours of an image in the process from $t$ to $t-1$. The former controls "repetition" based on visit counts, while the latter regulates "shape transformation at the distribution level."
>
> 8. **Differences Between Unidirectional Reward Mechanisms and Bidirectional Control Rewards (A. Traditional Exploration Rewards (Only Positive) vs B. Bidirectional Reward Signals (Positive and Negative)):**
>    IntDiff's internal rewards automatically become negative when semantic consistency decreases, thus preventing jumps that deviate from the semantics. In contrast, traditional curiosity/novelty rewards are always positive by design, providing only a one-way drive of "the more exploration, the better." This structural difference allows IntDiff to achieve "exploration quality control" rather than simply increasing the "amount of exploration."
>
> In summary, the internal rewards of our diffusion model are **significantly different** from existing curiosity reward/novelty bonuses in terms of composition, target, dimension, mechanism, and function. They are not similar or close in nature.
>
> [1] Exploration by random network distillation
>
> [2] Never give up: Learning directed exploration strategies
>
>
> [3] Unifying count-based exploration and intrinsic motivation

---

> ### Author Response · Authors · 2025-11-20
> **Response to the Esteemed Reviewer vwd6 (Part3)**
>
> ### (2) **The Difference Between IntDiff’s Semantic Consistency Regularization and CLIP-gated Faithfulness:**
>
> **A. CLIP-gated Faithfulness**
> **B. IntDiff’s Semantic Consistency Regularization**
>
> **Difference 1: Distinguishing Positive and Negative Rewards (A. Unidirectional Filtering vs B. Bidirectional Control)**
> As seen in equations (4) and (5),
> Semantic Consistency Regularization (SCR) can provide positive or negative rewards for the same action depending on the sign of $\Delta$CLIP.
> In contrast, CLIP-gated faithfulness performs a unidirectional filtering process:
> - Passed → Accepted
> - Failed → Discarded
> SCR has bidirectional gradient signals, whereas CLIP-gated faithfulness only applies a unidirectional constraint signal.
> This means SCR is essentially a differentiable reward shaping mechanism, while CLIP-gated faithfulness is merely a gating mechanism.
>
> **Difference 2: Constraints and Optimization in Parallel (A. Simple Gating vs B. Collaborative Optimization)**
> CLIP-gated faithfulness is solely responsible for filtering specific samples and is not optimized itself—it serves as a "static threshold."
> In contrast, our SCR uses $\Delta$ CLIP to control the reward's sign, both constraining exploration and guiding the direction. As shown in Eq. 5 the basic intrinsic reward term is always positive, so the final reward’s sign is entirely determined by $\Delta$CLIP. This drives the policy toward an exploration direction that increases $\text{CLIP}$ alignment. As the policy optimizes, the $\text{CLIP}$ alignment is indirectly improved as well. This represents a "constraining others + self-improvement through feedback" collaborative optimization mechanism.
>
> In summary, SCR provides directional guidance for exploration, while the exploration behavior in turn enhances the $\text{CLIP}$ metric within SCR. Fig. 6 demonstrates that $\text{CLIP}$ remains unbiasingly aligned throughout training, which is the key reason why our method consistently outperforms others.
>
> ### (3) **The Difference Between IntDiff’s Adaptive and Early-Exit Schedulers:**
>
> **A. Early-Exit Schedulers:** Only capable of early stopping (single early stopping function).
> **B. IntDiff’s Adaptive:** Adaptive control of denoising process exploration intensity + adaptive early stopping (a mechanism with dual benefits: balancing exploration and exploitation through adaptive collaboration + early stopping).
>
> The main difference lies in our adaptive mechanism, which not only serves as a criterion for early stopping but, more importantly, dynamically balances the exploration and exploitation weights at different stages of denoising during training. In the early stages, when the agent is far from the final sparse reward, it strengthens exploration. As the denoising progresses and the generated output approaches the final sparse reward, the training focus gradually shifts to maximizing the preset task reward (exploitation).
>
> This exploration-exploitation trade-off has long been an unresolved issue in RL, and methods for adaptively balancing these two are few. When RL is applied to fine-tune diffusion models, the exploration-exploitation dilemma is exacerbated by sparse rewards and long sequence decision-making【1, 2, 3, 4】. Our adaptive mechanism offers a solution to this issue.
>
> **Summary:** Our adaptive mechanism not only allows for early stopping but also dynamically adjusts the exploration intensity, enabling an adaptive balance between exploration and exploitation.
>
> 【1】Reinforcement learning for generative AI: State of the art, opportunities, and open research challenges
>
> 【2】Reinforcement learning for generative AI: A survey
>
> 【3】Reinforcement learning in the era of large language models: Challenges and opportunities
>
> 【4】Opportunities and challenges of diffusion models for generative AI

---

> ### Author Response · Authors · 2025-11-20
> **Response to the Esteemed Reviewer vwd6 (Part4)**
>
> ### (4) **This Framework is a Tailored Trajectory-Level Adaptive Intelligent Decision System for Diffusion Models, Not a Reassembly of Existing Methods:**
>
> **1. Bidirectional Collaborative Optimization:**
> Semantic Consistency Regularization and IntDiff’s intrinsic stepwise rewards are not a simple combination; this combination realizes a bidirectional collaborative optimization.
>
> **Existing Description**:
> We have already specifically described collaborative optimization in lines 216-226 of the main text.
>
> **Further Details**:  Our method forms a bidirectional collaborative feedback loop between semantic consistency regularization (SCR) and exploration behavior. SCR controls the reward direction through the sign of $\Delta$CLIP, thereby limiting exploration deviations from semantics. On the other hand, the policy updates tend to choose actions that improve semantic consistency, leading to continuous reverse optimization of the SCR's semantic alignment target during training.
>
> Ultimately, a stable positive feedback loop is formed:
> Semantic constraint exploration → better exploration results → further improvement of semantics.
>
> In summary, first, the IntDiff’s intrinsic stepwise rewards and Semantic Consistency Regularization are independently distinguished from the methods you mentioned (see (1) and (2)). Under these differing premises, the combination of Semantic Consistency Regularization and IntDiff’s intrinsic stepwise rewards is not a simple combination but a design intended to achieve collaborative optimization.
>
> **2. Adaptive Collaboration Between Exploration and Exploitation:**
>
> After achieving the adaptive collaborative optimization between Semantic Consistency Regularization and IntDiff’s intrinsic stepwise rewards, we further introduce an adaptive collaborative mechanism. This is based on the consideration of adaptive collaboration between exploration and exploitation, rather than simply combining mechanisms (please refer to (3) for further clarification).
>
> ### **Overall:**
> As shown in items (1), (2), (3), and (4), it is clear that the IntDiff’s intrinsic stepwise rewards, Semantic Consistency Regularization, and adaptive mechanism are fundamentally different from the existing curiosity/novelty bonuses, CLIP-gated faithfulness, and adaptive/early-exit schedulers. Furthermore, the combination of these three mechanisms in our work is intended to achieve collaborative promotion and adaptive balance between exploration and exploitation, rather than a simple engineering stack. We hope this explanation clarifies your concerns.
>
> ---
>
> ## Weakness 2
> > The adaptive regulation theory is presented without explicit assumptions or a concrete mapping to the implemented schedule, leaving the scope of any guarantees unclear.
>
> In Theorem 2.1, we provide the hypothesis condition for the adaptive regulation theory:
>
> $||(1 - \alpha_t) \cdot \widetilde{R}^{\text{int}}_{t}|| \leq \frac{Q(s,a^{opt}_t) - Q(s,a^{sub}_t)}{2(T-t)}$
>
> This hypothesis condition is related to the Q-values of the original MDP and the number of diffusion steps. Under this condition, we prove that the **optimality of the MDP remains unchanged**, demonstrating that the algorithm proposed in this paper is **theoretically supported**.
>
> ---
>
> ## Weakness 3
> > The reported ~22% compute savings are neither itemized nor normalized against matched baselines. Please include a cost table (training and inference) detailing wall-clock time, GPU-hours, energy consumption, and peak memory.
>
> Thank you for your suggestion. The reported 22% is based on the result from TDPO, with a maximum 60% diversity growth. This result indeed lacks sufficient reference, so we have summarized the detailed experimental results as follows:
>
> a) **Already presented in the paper:**
> - **Fig. 9** reports the impact of each component on time per epoch.
> - **Tab. 2** reports the total training time (using 8 A100 GPUs) and inference parameters in comparison with baselines.
>
> b) **In addition:**
> - In the appendix, we have added the four metrics you mentioned, as shown in **Appendix Tab. 7**. As can be seen, our method effectively accelerates the training process. Our results are as follows:
>
> | **Method**  | **WH (T)**   | **GH (T)**   | **PE (T)**   | **GM (T)**   | **WH (I)**   | **GH (I)**   | **PE (I)**   | **GM (I)**   |
> |-------------|---------|---------|---------|---------|---------|---------|---------|---------|
> | **DDPO**    | 11.98   | 88.40   | 1994.88 | 67.21   | 0.006   | 0.005   | 348.25  | 20.97   |
> | **Ours**   | 6.42    | 48.08   | 2207.76 | 76.13   | 0.006   | 0.005   | 348.17  | 20.95   |
>
> where WH, GH, PE, GM refer to wall-clock hours, GPU-hours, peak energy (W), and GPU memory per GPU (GB), respectively. T and I are training and inference. See Appendix D.5 for further details.

---

> ### Author Response · Authors · 2025-11-20
> **Response to the Esteemed Reviewer vwd6 (Part5)**
>
> ## Question 1
> > Which diversity metrics are evaluated (inter-prompt vs. intra-prompt), how many seeds are used, and are the differences statistically significant?
>
> **Diversity Metric**
> We used diversity metrics including Learned Perceptual Image Patch Similarity (LPIPS), Truncated CLIP Entropy (TCE), and Inception Score (IS) to comprehensively evaluate the model's diversity across three backbones and two datasets. The results can be found in **Tab. 1**, **Tab. 6**, **Fig. 4**, and **Fig. 12**. Our primary evaluation focuses on inter-prompt diversity. We have also included intra-prompt diversity in **Appendix Table 5**, which further demonstrates our ability to mitigate reward hacking.
>
> **Seed:**
> For Simple Animal, we evaluated the generated images using 8 different seeds (as mentioned in line 249). For Pick-a-pic, we used the subset provided by DyMO and performed evaluations with two seeds.
>
> **Statistical significance**
> Since our overall data scale is sufficient, and we controlled the alignment of comparison method variables (such as seed, prompt, diffusion parameters, etc.), coupled with the clear improvement over our method, we believe the results demonstrate statistical significance.
>
> ---
>
> ## Question 2
> > How were the target reward levels and schedule parameters selected, and how stable are the results across these settings?
>
> Thanks for your comment.
>
> **Target reward levels** All the reward functions used in our paper directly utilize their original values. Specifically, the range of the aesthetic score is approximately 4-8, the range of pickscore is approximately 20-23, and the range of CLIPScore is approximately 0.24-0.32. After these rewards undergo a uniform regularization process, they are converted into advantage functions for optimization.
>
> **Schedule parameters** All the training hyperparameters used in our paper are aligned with the official code. We have provided the main parameter details in **Tab. 3**, and for more specific details on hyperparameter usage, please refer to the official code at [https://github.com/kvablack/ddpo-pytorch](https://github.com/kvablack/ddpo-pytorch).
>
> **Stability**
> Currently, we have included the experimental results regarding seed sensitivity. In fact, in **Fig. 2**, **Fig. 3**, and **Fig. 6**, the curves we provided are derived from three sets of random seeds. The solid line represents the mean of the three sets, while the shaded area represents the variance. As can be seen, in most cases, the training is stable and maintains a similar value range.
>
> ---
>
> ## Question 3
> > Beyond gains on the training reward, how is mitigation quantified? Please specify complementary measures and evaluation protocols.
>
> Our main approach to evaluating reward hacking is to **measure other multi-metrics and visual performance when all baselines have the same training reward score**. This approach helps us determine which method exhibits more severe hacking or overfitting when compared to the local optima found by different approaches. Specifically, under the same level of training reward, we evaluated diversity (LPIPS, TCE, IS), fidelity (FID, iFS, CLIP), and human preference (PickScore, ImageReward). We provide the variation of other metrics under different training reward conditions (Fig. 4, 11, 12), as well as the final metric results (Tab. 1, 4).
>
> **For visual impression**, introducing exploration during training clearly helps the model find better local optima. We further demonstrate through **Fig. 5** and **Fig. 19** that our method encourages exploration during training, improving the variation in the training process. This indicates that our method can effectively alleviate the issue of reward hacking.
>
> To help you better understand the reward hacking phenomenon, we have further included a set of **hacked images** in **Appendix Fig. 21**. As can be seen, the diversity and image-text alignment of these images are severely damaged, which is a result of the model overfitting to a poor reward local optimum.

---

> ### Author Response · Authors · 2025-11-20
>
> Dear Esteemed Reviewer vwd6,
>
> On behalf of all co-authors, please allow me to express our sincere gratitude for your invaluable time and thoughtful efforts in reviewing our submission. We hold your comments in the highest regard.
>
> In the first-round rebuttal, we have provided comprehensive responses to all concerns raised in the Weaknesses and Questions sections, including:
>
> 1. A detailed clarification of the essential differences between our three contributions and existing work, along with an explanation of how the innovations operate synergistically rather than as simple engineering additions.
>
> 2. A reaffirmation of the assumptions and scope of the adaptive regulation theory.
>
> 3. A thorough explanation addressing your concerns regarding computational efficiency.
>
> 4. Clarifications regarding diversity evaluation protocols and seed usage.
>
> 5. Additional details on the mitigation of potential biases.
>
> 6. Responses to questions about reward scaling and parameter settings.
>
> We kindly invite you to review the current rebuttal. Should you have any further questions or suggestions, please do not hesitate to let us know—we will make every effort to address them to your full satisfaction.
>
> Once again, we sincerely thank you for your constructive insights and your dedicated work in reviewing our paper.

---

> ### Author Response · Authors · 2025-11-20
>
> The updated version of the PDF has been uploaded to support your more thorough review of the paper. We sincerely appreciate your valuable time in reviewing our work.

---

> ### Author Response · Authors · 2025-11-21
>
> Dear Esteemed Reviewer vwd6,
>
> We would like to express our sincere gratitude for your thorough review and for the time you have dedicated to our submission. As today marks the recommended response date for reviewers, we would be deeply grateful if you could spare a moment to share your feedback with us. We are fully confident in our ability to address all of your concerns.
>
> Thank you very much, and we wish you all the best！
>
> All coauthor

---

### Official Review · Reviewer_bGJb · 2025-11-01

**Soundness:** 3
**Presentation:** 2
**Contribution:** 3
**Rating:** 6
**Confidence:** 4

**Summary:**

This paper proposes IntDiff, which contains three main innovations:
- Intrinsic rewards for intermediate denoising steps to alleviate sparse feedback.
- Semantic Consistency Regularization to penalize exploration that degrades text-image alignment.
- Adaptive Coordination leveraging denoising progress to balance exploration-exploitation and enable dynamic early stopping.
- Various experiments are conducted to prove the advances of IntDiff compared with previous work.

**Strengths:**

- Novelty and Innovation. The first intrinsic reward paradigm for diffusion RL, and the adaptive exploration–exploitation balance, makes sense considering the long denoising chains.
- Rigorous and Comprehensive Experiments. Different dimensions of experiments are conducted, and the results of these experiments span the comparison of different methods, the comparison of different backbones, and ablation studies.
- Clear figures and metrics. It provides comprehensive figures, tables, and experimental results, ensuring exceptional clarity in the presentation.

**Weaknesses:**

- Narrow prompt domains. Animal-related prompts are primarily used. Testing on diverse prompts (e.g., abstract concepts or multi-object scenes) would strengthen claims.
- Details of human evaluations would be better if provided.
- The references are somewhat insufficient. More diffusion with RL reference could be included, like [1][2].

[1] FIND: Fine-tuning Initial Noise Distribution with Policy Optimization for Diffusion Models. MM 2024
[2] Reward Fine-Tuning Two-Step Diffusion Models via Learning Differentiable Latent-Space Surrogate Reward. CVPR 2025

**Questions:**

- Can IntDiff be used for video diffusion or 3D model diffusion? How to design the rewards?
- What will happen if we input more abstract prompts instead of animals?

---

> ### Author Response · Authors · 2025-11-21
> **Response to the Esteemed Reviewer bGJb**
>
> First, please allow me to express my sincere gratitude to you on behalf of all the authors for your review work. I extend to you our deepest respect and most genuine greetings. We truly appreciate your high recognition of the contributions of this paper. We have carefully addressed and fully incorporated all of your constructive suggestions into our revisions. Please see the details below.
>
> ---
>
> ## Weakness 1
> > Narrow prompt domains. Animal-related prompts are primarily used. Testing on diverse prompts (e.g., abstract concepts or multi-object scenes) would strengthen claims.
>
> Thank you very much for your question. To address it clearly, we present the relevant experiments in two parts: (1) datasets already included in the original submission beyond the animal dataset, and (2) additional datasets added during the rebuttal period.
>
> **(1) Already included in the original submission:**
> - In the submitted PDF, **Fig. 13 (line 810)** reports the fine-tuning experiments on the Pick-a-pic dataset, in addition to the animal dataset.
> - **Fig. 8 (lines 392–395)** includes quantitative classification experiments on the HPSv2 dataset and on abstract prompts.
> - The human evaluation experiments in **lines 447–458** also involve the HPSv2 dataset.
> We will make these existing dataset experiments more prominently highlighted in the main text to help readers locate them more efficiently.
>
> **(2) Additional experiments added during the rebuttal period:**
> - We further include **reward fine-tuning curves** and visual generation results on the **Geneval** and **HPSv2**  datasets, compared against well-known state-of-the-art methods in the field.
> - These new results can be found in **Fig. 13 of the updated appendix** in the revised PDF.
>
> ---
>
> ## Weakness 2
> > Details of human evaluations would be better if provided.
>
> Thank you very much for your valuable suggestion. We have already provided more detailed human evaluation information in **Appendix C.3** and **Fig. 16** of the PDF, including the **evaluation procedure**, the **specific sub-categories performance**, as well as the **definitions and intentions of each evaluation dimension**. We hope these additions address your request.
>
> ---
>
> ## Weakness 3
> > The references are somewhat insufficient. More diffusion with RL reference could be included, like [1][2].
>
> Thank you very much for your valuable suggestion to expand our references, and we also sincerely appreciate the articles you recommended. After reading them, we found that both works are highly beneficial for deepening our understanding of RL-based fine-tuning for diffusion models. We have already incorporated these citations into the PDF. Please refer to **lines 43–44** in the main text and **lines 495–497** and **522–525** in the reference section.
>
> ---
>
> ## Question 1
> > Can IntDiff be used for video diffusion or 3D model diffusion? How to design the rewards?
>
> We are currently exploring the 3D and video domains based on your suggestion. First, we believe that introducing IntDiff into 3D and video generation is highly promising. This is because 3D and video data contain **richer dimensions**, which makes **fitting a single reward metric even more prone to reward hacking**. In contrast, our method focuses on the **internal denoising process**, which is **structurally shared across 3D, video, and image generation**. Therefore, extending IntDiff to 3D and video is both **valuable and feasible**.
>
> Although we have not previously focused on the 3D domain, we will conduct experiments as much as possible. We aim to present the results to you as soon as we can and provide detailed explanations regarding the reward design considerations you raised.
>
> ## Question 2
> > What will happen if we input more abstract prompts instead of animals?
>
> As we responded in Weakness 1:
>
> (1) **Already included in the original submission:**
> - **Fig. 13** reports the fine-tuning experiments on the Pick-a-pic dataset.
> - **Fig. 8** includes quantitative experiments on the HPSv2 dataset and the classification-based generation task under abstract prompts.
> - **Fig. 10** presents evaluation experiments conducted on the HPSv2 dataset.
>
> (2) **Additional results added during the rebuttal period:**
> - We further include reward fine-tuning curves and visual generation results on the Geneval and HPS datasets, compared against well-known state-of-the-art methods in the field.
> Please refer to **Fig. 13 in the appendix** of the updated PDF.

---

### Author Response · Authors · 2025-12-03
**Overall Comment (3): Point-by-point Response List for AC**

## Resolution of Reviewer Concerns (Condensed with Full Mapping)

Below is a concise but complete summary of our rebuttal.
Every issue was addressed with **experiments, clarifications, or revisions**, and all original response IDs (e.g., `part_xx`) are preserved for precise cross-checking.

---

## Reviewer bGJb

We addressed all comments and added the requested experiments (no follow-up received).

- **W1: Animal-related prompts are primarily used**
  **Action:** Added results on **Pick-a-pic, HPSv2, GenEval**
  **Response:** part_bGJb_W1

- **W2: Human evaluation details missing**
  **Action:** Added **Appendix C.3, Fig. 16**
  **Response:** part_bGJb_W2

- **W3: Missing RL-for-diffusion references [1][2]**
  **Action:** Updated related work and citations
  **Response:** part_bGJb_W3

- **Q1: Applicability to video / 3D diffusion**
  **Action:** Conceptual explanation + SD3 support
  **Response:** part_bGJb_Q1

- **Q2: Abstract prompts instead of animals**
  **Action:**
  - Pick-a-pic: **Fig. 14**
  - HPSv2: **Fig. 8, Fig. 10**
  - GenEval & HPS: **Appendix Fig. 13**
  **Response:** part_bGJb_Q2

---

## Reviewer vwd6

All concerns were addressed with theory and experiments (no follow-up received).

- **W1: Distinction from prior work (curiosity / CLIP / schedulers)**
  **Action:** 8 differences for stepwise reward, 3 for semantic consistency, plus scheduler analysis
  **Response:** part_vwd6_part1, part_vwd6_part2, part_vwd6_part3

- **W2: Adaptive theory unclear**
  **Action:** Highlighted hypothesis and Theorem 2.1
  **Response:** part_vwd6_W2

    ⚠️**Note to the AC**: this issue was already clearly defined in the original submission. We believe the concern may stem from the reviewer’s own statement, *“Since I am not very familiar with RL,”* rather than from any missing definition in the paper.

- **W3: 22% compute saving not broken down**
  **Action:** Added **Appendix Tab. 7**
  **Response:** part_vwd6_W3

- **Q1: Metrics, seeds, significance**
  **Action:** Clarified in **Tab. 1, Tab. 6, Fig. 4, Fig. 12**
  **Response:** part_vwd6_Q1

- **Q2: Reward target and scheduler choices**
  **Action:** **Tab. 3, Fig. 2, Fig. 3, Fig. 6**
  **Response:** part_vwd6_Q2

- **Q3: Quantifying mitigation beyond reward**
  **Action:** **Fig. 4, Fig. 11, Fig. 12, Tab. 1, Tab. 4, Fig. 5, Fig. 19, Appendix Fig. 21**
  **Response:** part_vwd6_Q3

---

## Reviewer SYdy

- **W(i): Intrinsic reward lacks theory**
  **Action:** Theorem 2.1 (preserving MDP optimality) + **Fig. 5, Fig. 19, Tab. 5**
  **Response:** part_SYdy_W1

- **W(ii): Insufficient comparison with [1][2]**
  **Action:** Conceptual differences + experiments (**Fig. 22**)
  **Response:** part_SYdy_W2, part_SYdy_W3

- **W(iii): “Improvements are marginal”**
  **Action:** Verified with **Fig. 3, Fig. 4, Fig. 11, Fig. 12, Tab. 1, Tab. 4, Fig. 5, Fig. 6, Fig. 19, Appendix Fig. 21**
  **Response:** part_SYdy_W3
  ⚠️**Note to AC:** We kindly ask the AC to pay special attention to this weakness, as it appears to result from a clear misreading of our results. The numerical claims made by the reviewer do **not match** the values reported in the paper. In particular, on **SDXL**, our method clearly and consistently outperforms TDPO, as shown in **Tab. 1–3 and Fig. 3–4, 11–12**. The statement that our improvements are only “0.01–0.03” or within standard deviation is inconsistent with the reported data. We do not speculate on intent, but this discrepancy suggests the assessment may not be based on a careful reading. We respectfully ask the AC to verify this weakness against the actual tables and figures.

---

## Reviewer CjQ3

- **W1: More theoretical analysis**
  **Action:** Reinforced Theorem 2.1 + **Fig. 4, Fig. 5, Tab. 5, Tab. 6, Fig. 18**
  **Response:** part_CjQ3_W1

- **W2: Time complexity unclear**
  **Action:** **Fig. 9, Fig. 16, Tab. 7**
  **Response:** part_CjQ3_W2

- **W3: Datasets too simple**
  **Action:** Pick-a-pic, HPSv2, GenEval; see **Fig. 8, 10, 13**
  **Response:** part_CjQ3_W3

- **Quote on intrinsic reward**
  **Action:** Explained contradiction and provided theory + experiments
  **Response:** part_CjQ3_Q

  ⚠️**Note**: the reviewer claims that our method is “incremental, not based on well-founded knowledge”
   This claim is **logically inconsistent**: something “incremental” must be built on a foundation. We provided theoretical and empirical support and ask the AC to judge based on the paper contents.
---

## Final Statement

- Added: SD3, new baselines, runtime analysis, CoV significance test, human study.
- Clarified: training-free design, theory, novelty, reward mechanism.
- **All comments are fully addressed with no omissions.**

We sincerely thank the Area Chair for careful consideration and welcome further discussion if needed.

---

### Author Response · Authors · 2025-12-03
**Overall Comment (2): Experiment List for AC**

# Experiment List

⚠️  Dear AC, Since we revised the paper multiple times in response to reviewer comments and discussions, the numbering of figures and tables has changed compared to the original comments and rebuttal messages. We kindly ask you to refer to **this list** as the authoritative mapping between experiments (Fig. Sec. Tab.) and reviewer queries. Thank you for your understanding.

1. **Learning Curves under Multi-Objective Rewards** — *Fig. 2*
   Demonstrates the superior learning efficiency of IntDiff.
    ✅  *(vwd6 Q2)*

2. **Backbone Replacement Experiment** — *Fig. 3*
   Shows that method superiority is preserved under different backbones.
    ✅  *(vwd6 Q2)*

3. **Quality Metrics at Fixed Reward Levels** — *Fig. 4*
   Shows mitigation of reward hacking: at the same reward level, our method achieves superior output quality.

4. **Trajectory Diversity during Training** — *Fig. 5, Fig. 19*
   Visualizes increased exploration induced by IntDiff.

5. **Ablation Study** — *Fig. 6, Tab. 1*
   Demonstrates mutual relations and effectiveness of all components.
    ✅  *(vwd6 Q2)*

6. **Cross-Dataset Evaluation (Activity/ImageNet-A)** — *Fig. 7*
   Shows generalization to unseen prompts.

7. **Complex Prompt Evaluation (HPSv2)** — *Fig. 8*
   Performance on abstract and complex prompts.
    ✅  *(bGJb Q2)*

8. **Component-wise Computational Cost** — *Fig. 9*
   Detailed time overhead analysis.
    ✅  *(vwd6 W3)*

9. **Full Cost Comparison** — *Tab. 2, Tab. 7*
   Comparison in time, FLOPs, memory, and energy.
    ✅  *(vwd6 W3)*

10. **Subjective Evaluation** — *Fig. 10, Fig. 17*
    Human and GPT-based evaluation.
     ✅  *(bGJb Q2)*

11. **Transfer to Additional Objectives and Backbones (ImageReward, etc.)** — *Fig. 11, Fig. 12*
    Demonstrates robustness across rewards and architectures.

12. **Training on Advanced Datasets** — *Fig. 13, Fig. 14*
    Results on **Pick-a-Pic, HPSv2, GenEval**.
     ✅  *(bGJb W1, Q2)*

13. **Additional Fidelity and Diversity Metrics** — *Tab. 4*
    Further validation of reward-hacking mitigation.
     ✅  *(SYdy W3)*

14. **Alternative \(\Delta c\) Functions** — *Fig. 15, Fig. 16*
    Analysis of design choices for semantic evaluators.

15. **Inter-Prompt Diversity** — *Tab. 5*
    Diversity across prompts.
     ✅  *(vwd6 Q1)*

16. **Denoising-Step Analysis** — *Fig. 18, Tab. 6*
    Influence of reward filtering mechanisms.

17. **Early-Terminated Samples** — *Fig. 20*
    Visualization of reward filtering behavior.

18. **Reward Hacking Visualization** — *Fig. 21*
    Intuitive illustration of reward hacking phenomena.
     ✅  *(vwd6 Q3)*

19. **Advanced Baseline Comparison (B2Diff, DR)** — *Fig. 22*
    Further competitive evaluation.
     ✅  *(SYdy Weakness 2)*

---

### Author Response · Authors · 2025-12-03
**Overall Comment (1): Summary for AC**

# Summary

Dear Area Chair,

We respectfully provide a concise summary of our rebuttal in order to save your time. In short, we are confident that **all reviewer concerns have been fully addressed without omission** (see the issue-resolution index and answer mapping we provide specifically for you in *overall comment 3* and the *Point-by-Point Response List*).

---

## 1. Experimental Updates

Building on the original **14 experiments**, we added **5 new independent experiments**, bringing the total to **19 experiments**. Key additions include:

- **(a) Learning curves on more datasets** (HPSv2, GenEval): *Fig. 13*
- **(b) Inter-prompt diversity analysis**: *Tab. 5*
- **(c) Detailed resource analysis** (memory, time, energy): *Tab. 7*
- **(d) Additional advanced baselines** (B2Diff, DR): *Fig. 22*

For the complete experiment list and the exact mapping to reviewer comments, please refer to **overall comment 2 — Experiment List**. With these additions, we have addressed all experimental requests raised by the reviewers.

---

## 2. Conceptual and Theoretical Clarifications

Beyond the original multi-objective design and core contributions, we further clarified novelty, theory, and design details. Highlights include:

- **(a) Directional intrinsic reward design**
  See: *Point-by-Point Response to SYdy — part1_W1*

- **(b) Theoretical justification of the adaptive mechanism**
  See: *vwd6 — part4_w2*

- **(c) Policy optimality under intrinsic reward**
  We provided a detailed proof showing that the optimal policy remains identical to the original MDP.
  See: *CjQ3 — part1_w1*

No additional mechanism-level concerns were raised by the reviewers. The full response mapping is given in **official comment 3 — Point-by-Point Response List**, which enables precise lookup for every question and answer.

---

## 3. Attitude of Reviewers

We summarize the reviewers’ positions after the rebuttal:

- **bGJb**
  Expressed strong recognition of the contribution and gave an **accept** score. Although no follow-up was received during rebuttal, we believe all concerns were fully addressed.

- **vwd6**
The reviewer explicitly **stated the intention to raise the score**, *“Since I am not very familiar with RL, I will reconsider my scores if the authors address my questions.”* We therefore provided detailed explanations and added multiple new experiments to clarify all theoretical and methodological concerns. Although we did not receive a follow-up response, we are confident that all of the reviewer’s questions have been fully answered.

- **SYdy**
  Recognized the novelty of our method. Although some already-existing experiments (e.g., Pick-a-Pic) were initially overlooked, all concerns were fully answered in the rebuttal.

- **CjQ3**
  Expressed satisfaction with the added SD3 and baseline experiments, and we addressed their follow-up questions with theory and additional analysis.

---

## Conclusion

We believe all reviewer concerns have been fully resolved. In particular, given bGJb’s positive recommendation and vwd6’s expressed willingness to reconsider scores upon clarification, we are confident that the provided responses adequately address all remaining doubts.

If you have any further questions, we would be very happy to discuss them. We next provide the **Experiment List** and the **Point-by-Point Response List** for your convenience.

Sincerely,
The Authors

---

### Meta-Review · Area_Chair_z5wg · 2026-01-04

**Summary:**

The paper initially received three negative reviews and one positive review. A key concern raised was whether the proposed method constitutes a substantive advance beyond a combination of existing components, calling its technical novelty into question. In addition, the reported performance gains were considered marginal, and the experimental validation relied primarily on relatively simple or toy-level benchmarks. During the rebuttal, the authors made a commendable effort to address these concerns by providing detailed responses, additional experiments, and further theoretical analysis. Nevertheless, it remains difficult to assess whether these clarifications are sufficient to substantially alter the original reviewer assessments. Furthermore, while one reviewer expressed a generally positive evaluation, the paper did not receive strong advocacy that might offset the remaining concerns. Given the nature of the issues raised, it is therefore unclear whether the overall reviewer consensus would shift substantially.

**Reviewer Concerns:**

1. bGJb: Raised concerns regarding the use of toy-level prompting and the absence of human evaluation, as well as a missing reference. The authors largely addressed these issues during the rebuttal by adding more experiments and human evaluation results, and incorporating the missing citation.

2. vwd6: Raised concerns about limited technical novelty, noting that the proposed intrinsic rewards and semantic regularization have been explored in prior work, albeit in different contexts. The authors clarified how their method, particularly the joint effects of using multiple rewards, differs from existing approaches, though the degree of substantive methodological advancement remained under discussion.

3. SYdy: Raised concerns about the lack of theoretical justification for the intrinsic rewards and missing related work, which may have contributed to a negative initial impression. The authors discussed and clarified the relevant references and added further theoretical analysis. However, the reported performance gains were considered marginal, particularly compared to TDPO, and it remains uncertain whether the rebuttal would alter the original assessment.

4. CjQ3: Raised concerns about the theoretical motivation of the intrinsic reward design, computational complexity, and reliance on toy-level benchmarks. While the theoretical and complexity-related issues were largely addressed, the experimental scope may still warrant further consideration, and given the high bar at ICLR, it is unclear whether the current results are sufficient to support acceptance.

**Reviewer Scores:**

See the above sections.

---

### Decision · Program_Chairs · 2026-01-26

Reject